# Evolution of (p)ppGpp-HPRT regulation through diversification of an allosteric oligomeric interaction

Brent W Anderson[1], Kuanqing Liu[1], Christine Wolak[2], Katarzyna Dubiel[2], Fukang She[1], Kenneth A Satyshur[2], James L Keck[2], Jue D Wang[1]*

[1]Department of Bacteriology, University of Wisconsin, Madison, United States; [2]Department of Biomolecular Chemistry, University of Wisconsin, Madison, United States

**Abstract** The alarmone (p)ppGpp regulates diverse targets, yet its target specificity and evolution remain poorly understood. Here, we elucidate the mechanism by which basal (p)ppGpp inhibits the purine salvage enzyme HPRT by sharing a conserved motif with its substrate PRPP. Intriguingly, HPRT regulation by (p)ppGpp varies across organisms and correlates with HPRT oligomeric forms. (p)ppGpp-sensitive HPRT exists as a PRPP-bound dimer or an apo- and (p)ppGpp-bound tetramer, where a dimer-dimer interface triggers allosteric structural rearrangements to enhance (p)ppGpp inhibition. Loss of this oligomeric interface results in weakened (p)ppGpp regulation. Our results reveal an evolutionary principle whereby protein oligomerization allows evolutionary change to accumulate away from a conserved binding pocket to allosterically alter specificity of ligand interaction. This principle also explains how another (p)ppGpp target GMK is variably regulated across species. Since most ligands bind near protein interfaces, we propose that this principle extends to many other protein–ligand interactions.
DOI: https://doi.org/10.7554/eLife.47534.001

*For correspondence:
wang@bact.wisc.edu

Competing interests: The authors declare that no competing interests exist.

## Introduction

Regulation by signaling ligands is a universal mechanism for adaptation and homeostasis (*Chubukov et al., 2014*; *Traut, 2008*). How proteins evolve to be differentially regulated by their signaling ligands is under ongoing investigation (*Najmanovich, 2017*; *Taute et al., 2014*). A key signaling ligand in bacteria is the nucleotide (p)ppGpp. (p)ppGpp directly binds and regulates diverse protein targets at positions ranging from protein active sites to protein–protein interfaces (*Corrigan et al., 2016*; *Gourse et al., 2018*; *Liu et al., 2015a*; *Sherlock et al., 2018*; *Wang et al., 2019*; *Zhang et al., 2018*). However, how these diverse themes evolve across bacterial species have not been systematically elucidated. In addition, while most of the (p)ppGpp regulation is studied for high (p)ppGpp concentrations upon stress and starvation (*Cashel et al., 1996*; *Gourse et al., 2018*; *Liu et al., 2015a*), (p)ppGpp at basal levels also has important protective roles (*Gaca et al., 2013*; *Gaca et al., 2015a*; *Kriel et al., 2012*; *Potrykus et al., 2011*; *Puszynska and O'Shea, 2017*). However, molecular targets and mechanisms of basal (p)ppGpp regulation have not been delineated.

Here, we examined the specificity of (p)ppGpp binding by characterizing (p)ppGpp regulation of the enzyme hypoxanthine phosphoribosyltransferase (HPRT), one of the earliest identified targets of (p)ppGpp (*Hochstadt-Ozer and Cashel, 1972*; *Kriel et al., 2012*). Using biochemical, structural and evolutionary analyses of HPRT homologs across diverse bacterial phyla, we identify a conserved (p)ppGpp binding motif and identified a HPRT dimer–dimer interaction that allosterically positions a flexible loop to promote strong (p)ppGpp binding. The bacterial HPRT homologs lacking this dimer–dimer interaction are largely refractory to (p)ppGpp regulation. We propose that changes in protein

oligomerization enable evolution of regulation by (p)ppGpp, despite evolutionary constraint imposed on the direct (p)ppGpp-binding site. We examined this principle using another (p)ppGpp target GMK and discuss its broad implications in evolutionary diversification of signaling ligand specificity.

## Results

### Regulation of HPRT activity by basal levels of (p)ppGpp is important for GTP homeostasis

HPRT is a purine salvage enzyme that converts purine bases and PRPP to GMP and IMP, which are precursors of the essential nucleotide GTP (*Figure 1A*, upper panel). HPRT activity was previously found to be inhibited by (p)ppGpp in several bacteria including *Bacillus subtilis* (*Gaca et al., 2015b*; *Hochstadt-Ozer and Cashel, 1972*; *Kriel et al., 2012*). In *B. subtilis*, (p)ppGpp ranges from <30 µM at basal concentrations to millimolar levels upon starvation. (p)ppGpp inhibits *B. subtilis* HPRT activity at an $IC_{50}$ of $\approx$10 µM, allowing (p)ppGpp to regulate purine salvage at its basal concentrations (*Figure 1B*). The absence of (p)ppGpp in *B. subtilis* results in an uncontrolled GTP increase and toxicity upon guanine addition (*Kriel et al., 2012*) (*Figure 1A*, middle panel).

To demonstrate the physiological relevance of basal (p)ppGpp regulation of HPRT, we introduced the *E. coli* enzyme XGPRT (*ecgpt*) into *B. subtilis*. *E. coli* XGPRT produces GMP similarly to *B. subtilis* HPRT but is only modestly inhibited by (p)ppGpp at induced concentrations (*Figure 1B*). To visualize the effect of this regulation on GTP homeostasis, we also introduced a (p)ppGpp-resistant *E. coli* GMK to replace the (p)ppGpp-sensitive *B. subtilis* GMK, allowing the subsequent conversion of GMP to GDP to bypass (p)ppGpp regulation (*Figure 1A*, bottom panel). When these strains were treated with guanosine, GTP levels increased far more in cells with the modestly regulated *ecgpt* than in cells with the basal (p)ppGpp-regulated *B. subtilis hprT* (*Figure 1C*), demonstrating that basal regulation of HPRT by (p)ppGpp is important for preventing strong fluctuations in GTP.

We performed Michaelis Menten kinetics on *B. subtilis* HPRT and found (p)ppGpp is a strong competitive inhibitor of its substrate PRPP (*Figure 1D–E*). The $K_i$ for pppGpp is 1.7 µM, while the $K_m$ for PRPP is 166 µM, allowing (p)ppGpp at a basal level to effectively compete with physiological concentrations (0.1–1 mM) of PRPP (*Bennett et al., 2009*; *Hove-Jensen et al., 2017*) and keep GTP under surveillance not just during starvation, but also during unstarved growth.

### Conservation and variation of (p)ppGpp regulation of HPRT across species

It has been shown for the (p)ppGpp targets RNA polymerase and guanylate kinase that conservation of their binding to (p)ppGpp is limited within distinct bacterial phyla (*Liu et al., 2015b*; *Ross et al., 2016*). We therefore tested the conservation of HPRT regulation by conducting a broad biochemical survey of HPRT enzymes from free-living bacteria, human commensals, and pathogens. We revealed (p)ppGpp can bind and inhibit activities of HPRTs from every phylum of bacteria we tested, including Actinobacteria, Bacteroidetes, Firmicutes, Deinococcus-Thermus, and Proteobacteria, and even eukaryotic HPRTs (*Figure 2*). To do so, we purified 32 recombinant HPRTs and tested their enzymatic activities and found that most of them were strongly inhibited by ppGpp and pppGpp at 25 µM (*Figure 2A–B* and *Table 1*). Considering that (p)ppGpp is induced to millimolar concentrations during starvation and its uninduced levels are $\approx$10–30 µM during exponential growth in *B. subtilis* and *E. coli* (*Cashel et al., 1996*; *Kriel et al., 2012*), these results suggest significant inhibition of HPRT under physiological, basal levels of (p)ppGpp.

We also quantified the HPRT homologs' binding affinity ($K_d$) for pppGpp with <u>d</u>ifferential <u>r</u>adial <u>c</u>apillary <u>a</u>ction of <u>l</u>igand <u>a</u>ssay (DRaCALA) (*Figure 2C–D*) (*Roelofs et al., 2011*), results of which were highly comparable with well-established quantitative methods such as isothermal titration calorimetry (*Figure 2C* and *Figure 2—figure supplement 1*). Using this method, we found that most HPRTs bind pppGpp with $K_d$ values ranging from $\approx$0.1 to $\approx$10 µM (*Figure 2D* and *Table 1*).

Closer inspection revealed an additional quantitative difference in regulatory potency among these HPRTs. HPRTs from human microbiota commensals, including all species from Bacteroidetes along with the Firmicutes *Eubacterium* spp. and *Ruminococcus* spp., manifested the tightest interactions with pppGpp ($K_d \approx$ 0.1–1 µM) (*Figure 2D* and *Table 1*). This suggests a relationship between

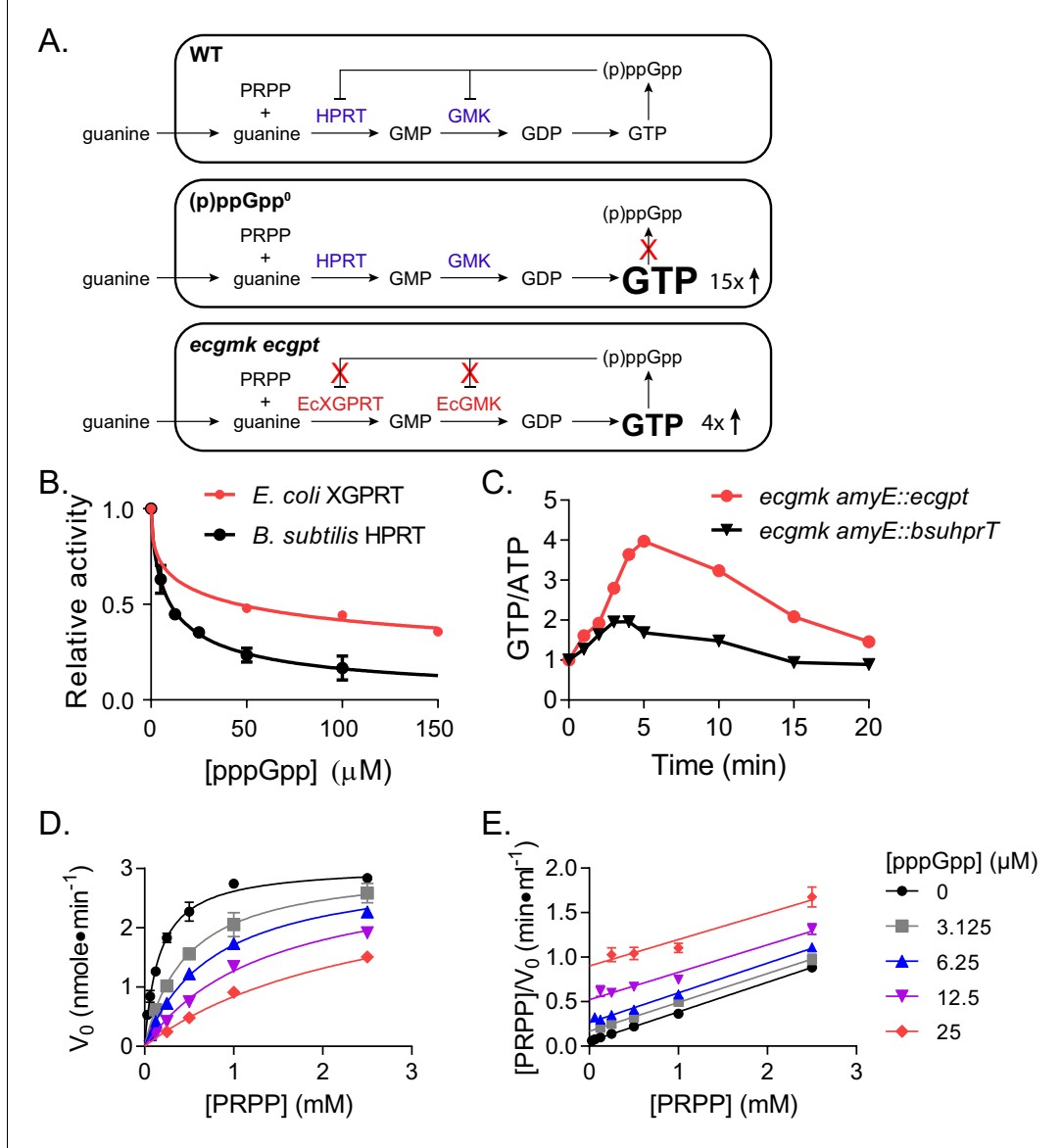

**Figure 1.** Regulation of HPRT by basal levels of (p)ppGpp is important for GTP homeostasis. (**A**) Pathways showing the effect of extracellular guanine on *B. subtilis* GTP homeostasis. In WT, (p)ppGpp regulates HPRT and GMK. (p)ppGpp⁰ cannot produce (p)ppGpp (see B) and *ecgmk ecgpt* has enzymes resistant to (p)ppGpp regulation (see F). (**B**) *E. coli* XGPRT more weakly inhibited by pppGpp than *B. subtilis* HPRT. The IC$_{50}$ for *E. coli* XGPRT is 45 µM compared to 10 µM for *B. subtilis* HPRT. Error bars represent SEM of triplicate. (**C**) Expression of *E. coli* XGPRT leads to imbalanced GTP/ATP homeostasis in *B. subtilis*. GTP/ATP ratio in *B. subtilis* treated with 1 mM guanosine as determined by thin layer chromatography of ³²P-labeled cells. Time is minutes after guanosine treatment. In *ecgmk*, (p)ppGpp-insensitive *E. coli gmk* replaces *B. subtilis gmk* at its endogenous locus. *ecgmk amyE:: bsuhprT* and *ecgmk amyE::ecgpt* express *B. subtilis hprT* and *E. coli gpt*, respectively, from an IPTG-dependent promoter at an exogenous locus. All strains grown with 1 mM IPTG. (**D**) Initial velocities of *B. subtilis* HPRT at varied PRPP and pppGpp concentrations. Data are fitted to a global competitive inhibition equation ($r^2$ = 0.975), and K$_i$ = 1.7 µM. (**E**) Data from (**D**) in a Hanes–Woolf transformation. Parallel lines indicate equal maximum velocities for each pppGpp concentration. Error bars represent SEM of at least three replicates.

DOI: https://doi.org/10.7554/eLife.47534.002

the intestinal environment and regulation of this purine salvage enzyme by (p)ppGpp, which may serve to buffer the intracellular environment against fluctuating extracellular purine concentrations. Beyond microbiota, HPRTs from soil-dwelling bacteria (e.g. *Streptomyces coelicolor*, *B. subtilis*) and pathogens (e.g. *Bacillus anthracis, Staphylococcus aureus, E. coli*), occupying the phyla Actinobacteria, Firmicutes and Proteobacteria, were also inhibited by (p)ppGpp with K$_d$ values below 10 µM.

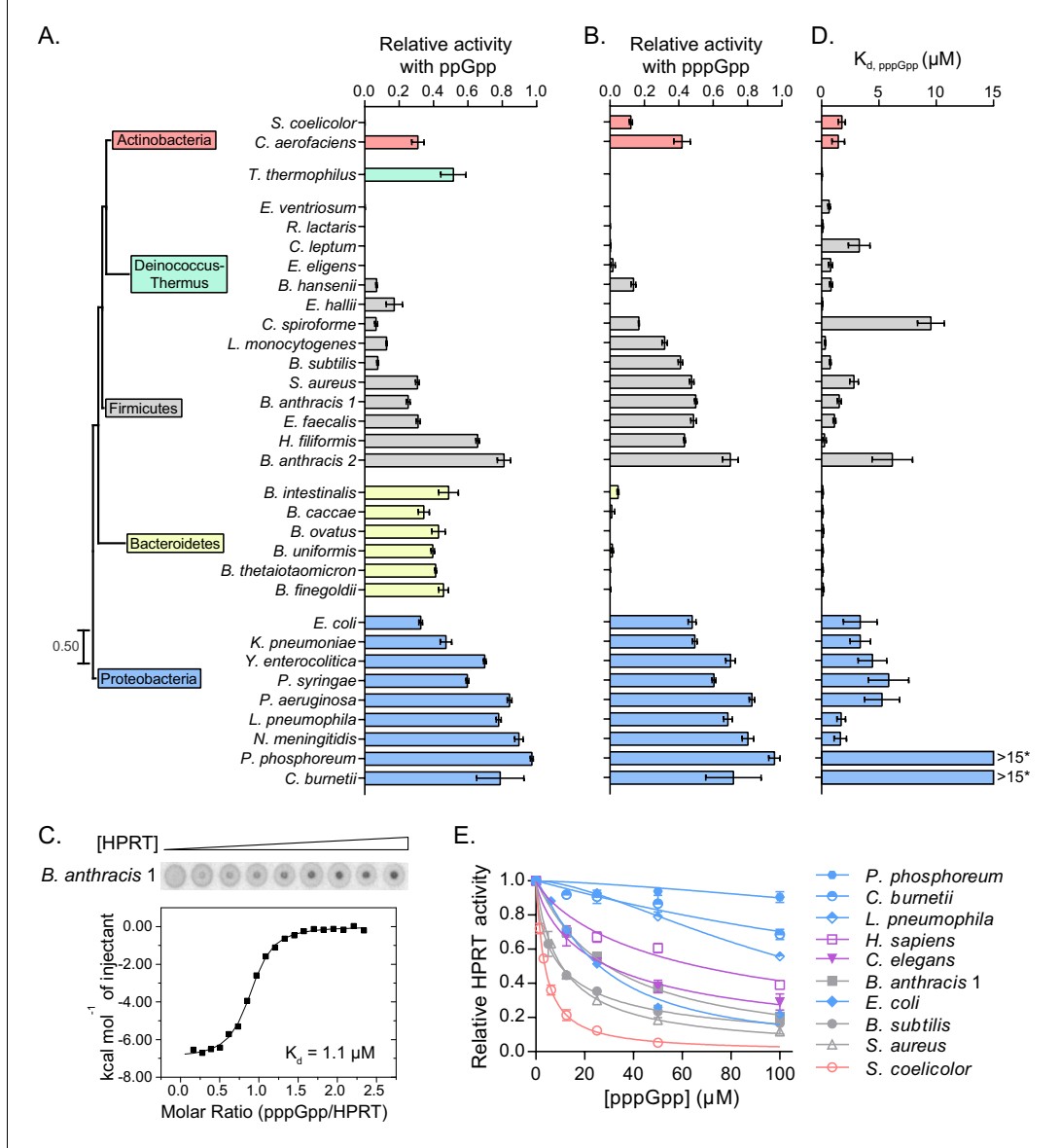

**Figure 2.** Conservation and variation in (p)ppGpp regulation of HPRTs across species. (A–B) (p)ppGpp inhibits HPRTs across species. Representative phylogenetic tree constructed from 16S rRNA sequences of species shown with eukaryotic outgroup hidden. Branch length scale represents 0.5 substitutions per site. See Materials and methods for tree construction. (A) Relative activities of HPRTs with 25 μM ppGpp. (B) Relative activities of HPRTs with 25 μM pppGpp. Error bars represent SEM of triplicates. (C–D) pppGpp binds HPRTs across species. (C) Representative DRaCALA between *B. anthracis* Hpt-1 in cell lysate serially diluted 1:2 and $^{32}$P-labeled pppGpp. The central signal is proportional to pppGpp – HPRT interaction. Binding isotherm from isothermal titration calorimetry between *B. anthracis* Hpt-1 and pppGpp. See *Figure 2—figure supplement 1* for energy isotherm and parameters. (D) The $K_d$ between pppGpp and HPRTs obtained with DRaCALA from serially diluted cell lysates containing overexpressed HPRTs (see Materials and methods). Error bars represent SEM derived from one binding curve. *P. phosphoreum* and *C. burnetii* HPRTs with * have affinities too weak to calculate but are estimated to be >15 μM (see *Figure 2—figure supplement 1*). (E) Relative activities of HPRTs with increasing concentrations of pppGpp. See *Table 1* for partial datasets for HPRTs from *M. tuberculosis*, *C. crescentus*, *S. meliloti*, *R. torques*, *S. mutans*, and *C. gilvus*.
DOI: https://doi.org/10.7554/eLife.47534.003

The following figure supplement is available for figure 2:

**Figure supplement 1.** Characterizing pppGpp interaction with bacterial HPRTs.
DOI: https://doi.org/10.7554/eLife.47534.004

**Table 1.** (p)ppGpp binding and inhibition of HPRT homologs.

| Organism | Relative activity[d] | | $K_{d\ (pppGpp)}$ (µM) | $IC_{50\ (pppGpp)}$ (µM) |
|---|---|---|---|---|
| | 25 µM ppGpp | 25 µM pppGpp | | |
| **Actinobacteria** | | | | |
| Streptomyces coelicolor | −0.01 ± 0.01 | 0.12 ± 0.01 | 1.77 ± 0.31 | 3.8 ± 0.2 |
| Collinsella aerofaciens | 0.31 ± 0.04 | 0.42 ± 0.05 | 1.46 ± 0.56 | |
| Cellulomonas gilvus | NA[b] | NA[b] | 0.6 ± 0.11 | |
| Mycobacterium tuberculosis | NA[a] | NA[a] | NA[c] | |
| **Bacteroidetes** | | | | |
| Bacteroides intestinalis | 0.49 ± 0.06 | 0.05 ± 0.01 | 0.11 ± 0.03 | |
| Bacteroides caccae | 0.34 ± 0.03 | 0.01 ± 0.02 | 0.11 ± 0.02 | |
| Bacteroides ovatus | 0.43 ± 0.04 | −0.03 ± 0.02 | 0.15 ± 0.04 | |
| Bacteroides uniformis | 0.4 ± 0.01 | 0.01 ± 0.01 | 0.1 ± 0.04 | |
| Bacteroides thetaiotaomicron | 0.41 ± 0 | 0 ± 0.01 | 0.11 ± 0.04 | |
| Bacteroides finegoldii | 0.46 ± 0.03 | −0.01 ± 0.01 | 0.15 ± 0.06 | |
| **Firmicutes** | | | | |
| Eubacterium ventriosum | −0.03 ± 0.03 | −0.07 ± 0.03 | 0.66 ± 0.1 | |
| Ruminococcus lactaris | −0.01 ± 0 | 0 ± 0 | 0.11 ± 0.03 | |
| Ruminococcus torques | NA[b] | NA[b] | 0.22 ± 0.07 | |
| Clostridium leptum | −0.01 ± 0.01 | 0 ± 0 | 3.29 ± 0.94 | |
| Eubacterium eligens | −0.03 ± 0.03 | 0.02 ± 0.01 | 0.79 ± 0.18 | |
| Blautia hansenii | 0.07 ± 0 | 0.14 ± 0.01 | 0.83 ± 0.14 | |
| Eubacterium hallii | 0.17 ± 0.05 | −0.06 ± 0.05 | 0.07 ± 0.02 | |
| Clostridium spiroforme | 0.07 ± 0.01 | 0.17 ± 0 | 9.54 ± 1.17 | |
| Listeria monocytogenes | 0.13 ± 0 | 0.32 ± 0.02 | 0.32 ± 0.06 | |
| Bacillus subtilis | 0.08 ± 0.01 | 0.41 ± 0.01 | 0.75 ± 0.07 | 10.1 ± 1.3 |
| Staphylococcus aureus | 0.31 ± 0.01 | 0.48 ± 0.01 | 2.83 ± 0.38 | 10.6 ± 0.3 |
| Streptococcus mutans | NA[b] | NA[b] | 5.12 ± 1.45 | |
| Bacillus anthracis 1 | 0.25 ± 0.01 | 0.5 ± 0.01 | 1.54 ± 0.18 | 29.5 ± 1.2 |
| Enterococcus faecalis | 0.31 ± 0.01 | 0.49 ± 0.02 | 1.12 ± 0.11 | |
| Holdemania filiformis | 0.66 ± 0.01 | 0.43 ± 0 | 0.17 ± 0.11 | |
| Bacillus anthracis 2 | 0.81 ± 0.04 | 0.7 ± 0.05 | 6.18 ± 1.77 | |
| **Deinococcus-Thermus** | | | | |
| Thermus thermophilus | 0.52 ± 0.07 | −0.01 ± 0.01 | 0.04 ± 0.01 | |
| **Proteobacteria** | | | | |
| Caulobacter crescentus | 0.33 ± 0.05 | 0.47 ± 0.07 | 0.24 ± 0.09 | |
| Escherichia coli | 0.33 ± 0.01 | 0.48 ± 0.02 | 3.38 ± 1.47 | 26.2 ± 2.0 |
| Klebsiella pneumoniae | 0.47 ± 0.03 | 0.49 ± 0.01 | 3.38 ± 0.89 | |
| Sinorhizobium meliloti | NA[b] | NA[b] | 3.08 ± 0.92 | |
| Yersinia enterocolitica | 0.7 ± 0.01 | 0.7 ± 0.03 | 4.44 ± 1.25 | |
| Photobacterium phosphoreum | 0.97 ± 0.01 | 0.96 ± 0.03 | NA[c] | >100 |
| Pseudomonas syringae | 0.6 ± 0.01 | 0.6 ± 0.01 | 5.85 ± 1.76 | |
| Pseudomonas aeruginosa | 0.84 ± 0.01 | 0.83 ± 0.02 | 5.28 ± 1.52 | |
| Legionella pneumophila | 0.78 ± 0.01 | 0.69 ± 0.03 | 1.72 ± 0.38 | >100 |
| Coxiella burnetii | 0.79 ± 0.14 | 0.72 ± 0.16 | NA[c] | >100 |
| Neisseria meningitidis | 0.9 ± 0.02 | 0.8 ± 0.03 | 1.64 ± 0.55 | |

*Table 1 continued on next page*

*Table 1 continued*

| Organism | Relative activity[d] | | $K_{d\ (pppGpp)}$ (µM) | $IC_{50\ (pppGpp)}$ (µM) |
|---|---|---|---|---|
| | 25 µM ppGpp | 25 µM pppGpp | | |
| Eukarya | | | | |
| *Saccharomyces cerevisiae* | 0.64 ± 0.02 | 0.96 ± 0.01 | 8.08 ± 8.58 | |
| *Caenorhabditis elegans* | NA[b] | NA[b] | 12.77 ± 7.68 | 29.6 ± 2.8 |
| *Homo sapiens* | NA | NA | NA | 66.6 ± 5.4 |

± standard error of the mean.

NA = no data obtained.

[a]Protein not purified.

[b]Protein did not have activity.

[c]Interaction with pppGpp too weak to estimate $K_d$.

[d]Relative to activity without ppGpp or pppGpp.

DOI: https://doi.org/10.7554/eLife.47534.005

Interestingly, a few HPRTs were only weakly inhibited by 25 µM (p)ppGpp (*Figure 2A–B*). These weakly regulated HPRTs all clustered in β- and γ-Proteobacteria (e.g. *Pseudomonas aeruginosa* and *Neisseria meningitidis*), with the exception of *Mycobacterium tuberculosis* (*Figure 2—figure supplement 1C*). To test whether the HPRTs weakly inhibited at basal levels of (p)ppGpp (25 µM) can be inhibited by higher levels of (p)ppGpp, we examined select HPRT homologs at a range of pppGpp concentrations and at physiological substrate concentrations (1 mM PRPP and 50 µM guanine) (*Bennett et al., 2009*) (*Figure 2E*). Most bacterial HPRTs (e.g. *B. subtilis*, *E. coli*, *S. aureus*) were sensitive to basal pppGpp. However, several bacterial HPRTs that were weakly inhibited by 25 µM pppGpp (e.g. *C. burnetii*, *L. pneumophila*) are also almost refractory to high concentration of pppGpp. In fact, they are less inhibited by pppGpp than eukaryotic HPRTs from human and *C. elegans*, for which (p)ppGpp is not a physiological regulatory ligand. Therefore, we can broadly classify bacterial HPRTs by their sensitivity to (p)ppGpp: those that are sensitive to basal (p)ppGpp and a few that are resistant to basal (p)ppGpp, although may still be mildly sensitive to induced (p)ppGpp.

In addition to differences between HPRT homologs, we also noticed a strong, species-dependent difference between pppGpp and ppGpp inhibition. All HPRTs in Bacteroidetes were potently inhibited by pppGpp but only weakly by ppGpp (*Figure 2A–B*). In contrast, nearly all other HPRTs displayed stronger inhibition by ppGpp than pppGpp. It is noted that *Bacteroides spp.*, whose HPRTs are more sensitive to pppGpp than ppGpp, contain a GppA homolog, and they make mostly ppGpp in vivo (*Glass et al., 1979*). On the other hand, Firmicutes, whose HPRTs are more sensitive to ppGpp than pppGpp, lack GppA homologs and produce predominantly pppGpp. Therefore, although pppGpp and ppGpp are often regarded as similar, they can have marked differences for certain cellular targets.

## (p)ppGpp binds the conserved active site of HPRT and closely mimics substrate binding

To examine the molecular determinants underlying (p)ppGpp regulation of HPRT, we turned to Hpt-1 from the pathogenic bacterium *B. anthracis*, which pppGpp competitively inhibits similarly to *B. subtilis* HPRT (*Figure 3—figure supplement 1*). We crystallized Hpt-1 with and without ppGpp (*Figure 3A* and *Table 2*), and for comparison, we also crystallized Hpt-1 with its two substrates, PRPP and the non-reactive guanine analog 9-deazaguanine (*Figure 3B* and *Table 2*) (*Héroux et al., 2000*). Apo HPRT diffracted to 2.06 Å resolution with four molecules in the asymmetric unit and was nearly identical to a deposited apo structure of *B. anthracis* Hpt-1 (PDB ID 3H83) (*Figure 3—figure supplement 2*). The HPRT-substrates structure diffracted to 1.64 Å containing two molecules in the asymmetric unit (*Figure 3—figure supplement 2*), and PRPP and 9-deazaguanine were coordinated by two $Mg^{2+}$ ions with each monomer (*Figure 3B* and *Figure 3—figure supplement 3*). The HPRT-ppGpp complex diffracted to 2.1 Å resolution with two molecules in the asymmetric unit (*Figure 3—figure supplement 2*), and each monomer contained one ppGpp coordinated with $Mg^{2+}$, in agreement with near 1:1 stoichiometry measured via ITC (*Figure 2—figure supplement 1* and *Figure 3—*

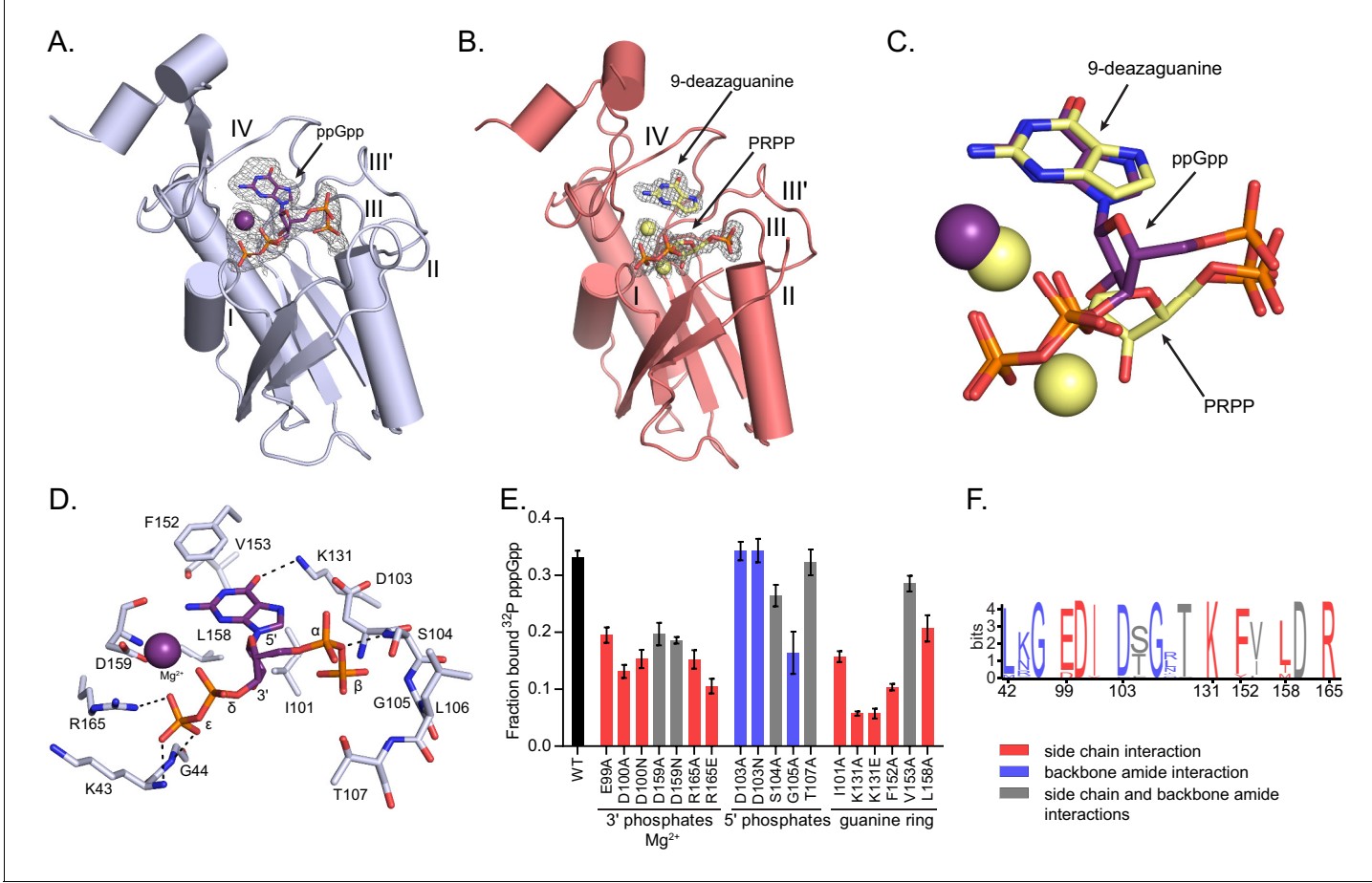

**Figure 3.** (p)ppGpp binds the HPRT active site and shares an almost identical binding pocket with substrates. (**A**) *B. anthracis* Hpt-1 crystallized with ppGpp. Five loops (I, II, III, III', IV) form the HPRT active site (*Sinha and Smith, 2001*). ppGpp and $Mg^{2+}$ shown with omit electron density contoured at 2σ. *Figure 3—figure supplement 1* shows that pppGpp competitively inhibits Hpt-1. *Figure 3—figure supplement 4* shows omit electron density for the two ppGpp molecules crystallized in the asymmetric unit. (**B**) *B. anthracis* Hpt-1 crystallized with 9-deazaguanine, PRPP, and two $Mg^{2+}$. Ligands shown with omit electron density contoured at 2.5σ. Residues Tyr70 – Ser76 in loop II were not resolved. See *Figure 3—figure supplement 2* for asymmetric units of Hpt-1 crystallized with ppGpp, substrates, and sulfates. See *Figure 3—figure supplement 3* for coordination of $Mg^{2+}$ with ppGpp and PRPP. (**C**) Overlay of substrates (PRPP and 9-deazaguanine; yellow) and inhibitor (ppGpp; purple) bound to HPRT. Spheres represent $Mg^{2+}$ and are colored according to their coordinating ligand. See *Figure 3—figure supplement 6* for binding pocket comparison. (**D**) The ppGpp binding pocket on HPRT. Black dotted lines indicate select hydrogen bonds. The peptide backbone is shown for residues where the interactions are relevant. See *Figure 3—figure supplement 5* for complete interaction maps. (**E**) DRaCALA of *B. subtilis* HPRT variants binding to $^{32}P$-labeled pppGpp, determined with cell lysates containing overexpressed HPRT variants. Disruption of side chain interactions weakened binding. For E and F, the residues are colored according to their interaction with ppGpp. Red = side chain interaction, blue = backbone amide interaction, and gray = both side chain and backbone interactions. Error bars represent SEM of three replicates. (**F**) Sequence frequency logo of the (p)ppGpp binding pocket from 99 bacterial HPRTs. Logo created using WebLogo from UC-Berkeley. See *Figure 3—figure supplement 8* for binding residues from select bacterial and eukaryotic HPRTs.
DOI: https://doi.org/10.7554/eLife.47534.006

The following figure supplements are available for figure 3:

**Figure supplement 1.** pppGpp competes with PRPP to inhibit *B. anthracis* Hpt-1.
DOI: https://doi.org/10.7554/eLife.47534.007
**Figure supplement 2.** Asymmetric units of *B.anthracis* Hpt-1 structures.
DOI: https://doi.org/10.7554/eLife.47534.008
**Figure supplement 3.** Coordination of $Mg^{2+}$ in *B. anthracis* Hpt-1 crystallized with ppGpp and substrates.
DOI: https://doi.org/10.7554/eLife.47534.009
**Figure supplement 4.** Omit electron densities of ppGpp crystallized with *B. anthracis* Hpt-1.
DOI: https://doi.org/10.7554/eLife.47534.010
**Figure supplement 5.** Primary structure of *B. anthracis* Hpt-1 showing ligand-interacting residues.
DOI: https://doi.org/10.7554/eLife.47534.011
*Figure 3 continued on next page*

*Figure 3 continued*

**Figure supplement 6.** Comparison between ppGpp-bound and substrates-bound binding pocket.
DOI: https://doi.org/10.7554/eLife.47534.012
**Figure supplement 7.** LigPlots of ppGpp crystallized with *B. anthracis* Hpt-1.
DOI: https://doi.org/10.7554/eLife.47534.013
**Figure supplement 8.** Conservation of the ppGpp-binding site across select bacteria and eukaryotes.
DOI: https://doi.org/10.7554/eLife.47534.014

*figure supplement 3*). These crystals formed in drops containing pppGpp, but there was insufficient density to completely model the 5′ γ-phosphate (*Figure 3—figure supplement 4*). With LC-MS/MS, we verified that the pppGpp was not contaminated with ppGpp. Although it is possible that the γ-phosphate was hydrolyzed during crystallization, the presence of waters and unassigned density around the 5′ phosphates allow for the possibility that the γ-phosphate is present but dynamic.

Comparison between the HPRT-ppGpp and HPRT-substrates structures revealed that ppGpp binds the HPRT active site and closely mimics the conformation of the two substrates (*Figure 3C* and *Figure 3—figure supplements 5* and *6*). The purine ring of ppGpp overlaps with the purine base substrate, and the two phosphate arms of ppGpp and PRPP both spread across the active site between loop I and loop III (*Figure 3C* and *Figure 3—figure supplement 6*). Generally speaking, in ppGpp–protein interactions, the phosphates of ppGpp are either elongated or compacted in a ring-like conformation (*Steinchen and Bange, 2016*). With the phosphates of ppGpp spread across the binding pocket, the HPRT-ppGpp interaction represents an elongated conformation (*Figure 3A*).

The (p)ppGpp-binding site in HPRT manifests a novel (p)ppGpp–protein interaction. The 5′ phosphates and ribose of ppGpp interact with loop III (EDIIDSGLT), a well-characterized PRPP binding motif (*Sinha and Smith, 2001*), through side chain interactions with Glu99 and Asp100 and backbone amide interactions with Asp103 – Thr107 (*Figure 3D* and *Figure 3—figure supplement 7*). The 3′ phosphates of ppGpp are coordinated by backbone amides of loop I, the side chain of Arg165, and the Mg$^{2+}$ ion (*Figure 3D* and *Figure 3—figure supplement 7*). The guanine ring of ppGpp is surrounded by a hydrophobic cleft formed by Ile101, Phe152, and Leu158, and Lys131 hydrogen bonds the guanine's exocyclic oxygen (*Figure 3D* and *Figure 3—figure supplement 7*). We validated the (p)ppGpp-binding residues by mutation analyses which showed that altering residues with side chain interactions with (p)ppGpp greatly weakened pppGpp binding (*Figure 3E*). In sum, this binding pocket illustrates the first example of a (p)ppGpp binding motif that shares the PRPP-binding motif. Since many proteins bind PRPP (*Hove-Jensen et al., 2017*), this (p)ppGpp motif may represent a new class of (p)ppGpp targets.

A frequency logo of the binding pocket from 99 bacterial HPRTs shows that most (p)ppGpp-interacting residues are highly conserved across bacteria (*Figure 3F* and *Figure 3—figure supplement 8*), and nearly all binding residues are also conserved in the eukaryotic HPRTs (*Figure 3—figure supplement 8*). The strong conservation across species is not surprising given the close overlap between (p)ppGpp and substrates in the HPRT structures. Since all residues involved in binding ppGpp are also involved in binding substrates, altering the site to affect inhibitor binding would also impact enzyme activity.

The nearly identical recognition of (p)ppGpp and substrates, along with the conservation of the active site, raised the following question: how can some HPRTs be strongly inhibited by basal levels of (p)ppGpp and other HPRTs be almost refractory to (p)ppGpp control despite sharing a conserved binding pocket?

## (p)ppGpp prevents PRPP-induced dissociation of HPRT dimer-of-dimers

We noticed a significant difference between the ppGpp- and substrates-bound tertiary structures in the conformation of a flexible loop. This loop, also called loop II, is common to all HPRTs and covers the active site during catalysis (*Shi et al., 1999*). In the HPRT-ppGpp complex, one side of loop II faces the active site while the other side is a critical part of the interface between two HPRT dimers (*Figure 4A*). We applied PISA analysis ('Protein interfaces, surfaces, and assemblies' service at European Bioinformatics Institute) to predict that this dimer-dimer interaction results in a tetrameric HPRT (*Krissinel and Henrick, 2007*). In the substrates-bound state, loop II is instead shifted ≈4 Å toward the active site from its position in the dimer-dimer interface (*Figure 4A*). Loop II also pulls its

**Table 2.** X-ray data collection and structure determination statistics.

**Data collection**

| Structure | HPRT with sulfates (PDB 6D9Q) | HPRT with substrates (PDB 6D9R) | HPRT with ppGpp (PDB 6D9S) |
|---|---|---|---|
| Wavelength | 0.9786 | 0.97872 | 0.9786 |
| Resolution range (highest resolution bin) (Å) | 41.3–2.06 (2.13–2.06) | 50–1.64 (1.67–1.64) | 40.2–2.11 (2.18–2.11) |
| Space group | P $3_1$ 2 1 | P $3_2$ 2 1 | P $3_1$ 2 1 |
| Unit cell | | | |
| a, b, c (Å) | 82.597, 82.597, 242.416 | 113.758, 113.758, 56.731 | 82.61, 82.61, 174.92 |
| $\alpha, \beta, \gamma$ (°) | 90, 90, 120 | 90, 90, 120 | 90, 90, 120 |
| Completeness (%) | 99.16 (95.52) | 99.8 (99.0) | 98.92 (94.51) |
| Unique reflections | 60264 | 51750 | 40413 |
| Redundancy | 10.8 (6.4) | 21.5 (14.4) | 8.1 (6.2) |
| $I/\sigma I$ | 24.8 (1.74) | 31.09 (2.93) | 16.41 (1.22) |
| $R_{meas}$ | 0.124 | 0.137 (1.376) | 0.164 |
| $R_{pim}$ | 0.037 | 0.029 (0.35) | 0.06 |
| CC 1/2 | (0.791) | (0.81) | (0.724) |

**Refinement**

| | | | |
|---|---|---|---|
| Resolution range (highest resolution bin) (Å) | 41.3–2.06 (2.11–2.06) | 37.19–1.64 (1.70–1.64) | 40.2–2.11 (2.16–2.11) |
| $R_{work}/R_{free}$ [a] (%) | 18.2/21.7 | 16.5/19.6 | 20.2/24.4 |
| r.m.s.[b] deviations | | | |
| Bonds (Å) | 0.004 | 0.01 | 0.003 |
| Angles (Å) | 1.03 | 1.41 | 0.812 |
| Ramachandran statistics (%) | | | |
| Favored | 97.90 | 98.52 | 96.63 |
| Allowed | 2.10 | 1.48 | 3.37 |
| Disallowed | 0.00 | 0.00 | 0.00 |
| Rotamer outliers (%) | 0.78 | 0.30 | 0.63 |
| No. of atoms | | | |
| Macromolecules | 5742 | 2881 | 2837 |
| Ligands | 46 | 161 | 116 |
| Solvent | 440 | 317 | 118 |
| B factor (Å$^2$) | | | |
| Macromolecules | 50.63 | 24.06 | 76.03 |
| Ligands | 68.83 | 37.76 | 161.17 |
| Solvent | 52.15 | 35.53 | 70.22 |

[a]$R_{work}/R_{free} = \Sigma ||F_{obs}| - |F_{calc}||/|F_{obs}|$, where the working the free R factors are calculated by using the working and free reflection sets, respectively. The free R reflections were held aside throughout refinement. [b]Root mean square. See **Table 2—source datas 1–3** for PDB validation reports.

DOI: https://doi.org/10.7554/eLife.47534.015

The following source data is available for Table 2:
**Source data 1.** Validation report for PDB ID 6D9Q.
DOI: https://doi.org/10.7554/eLife.47534.016
**Source data 2.** Validation report for PDB ID 6D9R.
DOI: https://doi.org/10.7554/eLife.47534.017
**Source data 3.** Validation report for PDB ID 6D9S.
DOI: https://doi.org/10.7554/eLife.47534.018

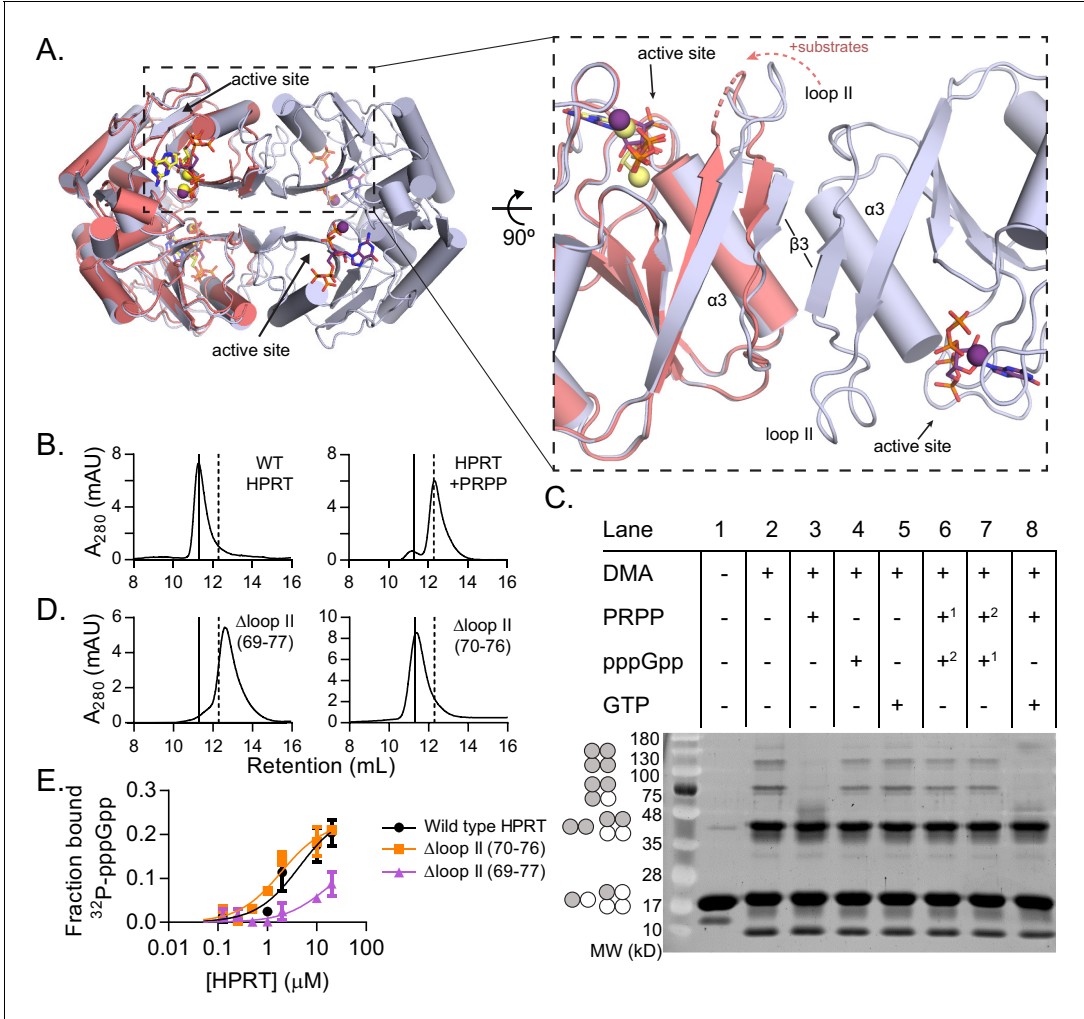

**Figure 4.** (p)ppGpp counteracts substrate-induced HPRT dimerization and HPRT tetramerization potentiates (p)ppGpp binding. (A) Overlay of HPRT tetramer crystallized with ppGpp (silver) and HPRT dimer crystallized with substrates (salmon). ppGpp is purple and the substrates are yellow. Inset, View of the dimer–dimer interface within an HPRT tetramer. Secondary structure components at the interface are labeled β3, α3, and loop II. *Figure 4—figure supplement 1* shows that changes in loop II are not induced by crystal contacts. (B) Size-exclusion chromatography of *B. subtilis* HPRT without ligand (left) and with PRPP (right). PRPP addition shifts the oligomeric state from tetramer to dimer. *B. anthracis* Hpt-1 is also a dimer with PRPP (see *Figure 4—figure supplement 2*). In B and D, solid line shows *B. subtilis* HPRT tetramer peak and dotted line shows dimer peak. PRPP must be in mobile phase to cause the shift (*Figure 4—figure supplement 3*), and the purine base does not affect the oligomeric state with PRPP (*Figure 4—figure supplement 4*). For molecular weight standards, see *Figure 4—figure supplement 7*. (C) *B. subtilis* HPRT crosslinked with dimethyl adipimidate (DMA). Crosslinked multimers were separated on SDS–PAGE. Predicted multimers are shown on the left. Shaded circles represent predicted to be crosslinked monomers, and white circles represent incomplete crosslinking. [1, 2] indicates order of incubation. GTP competition with PRPP did not recapitulate pppGpp blockage of PRPP-induced dimer–dimer dissociation which could be due to difference in affinity, whereby pppGpp binds strongly enough to outcompete PRPP but GTP does not. See *Figure 4—figure supplement 5* for effect of crosslinked tetramer (Y117C) on HPRT activity and PRPP competition with pppGpp. See *Figure 4—figure supplement 6* for ITC with PRPP. (D) Size-exclusion chromatography of *B. subtilis* HPRT with a partial loop II deletion (Δ70–76) or a complete loop II deletion (Δ 69–77). The complete deletion results in dimerization of the protein. (E) Binding curves between $^{32}$P-labeled pppGpp and *B. subtilis* HPRT variants obtained with DRaCALA. The tetrameric Δloop II (70-76) variant (orange squares) binds as well as wild type HPRT. The dimeric Δloop II (69-77) variant (purple triangles) displays weaker binding. Error bars represent SEM of three replicates.

DOI: https://doi.org/10.7554/eLife.47534.019

The following figure supplements are available for figure 4:

**Figure supplement 1.** Crystal contacts do not induce changes in loop II in *B. anthracis* Hpt-1 crystallized with substrates.

DOI: https://doi.org/10.7554/eLife.47534.020

**Figure supplement 2.** *B. anthracis* Hpt-1 is a tetramer without ligands and a dimer with PRPP.

DOI: https://doi.org/10.7554/eLife.47534.021

**Figure supplement 3.** PRPP does not cause HPRT dimerization when not in the mobile phase.

*Figure 4 continued on next page*

*Figure 4 continued*

DOI: https://doi.org/10.7554/eLife.47534.022

**Figure supplement 4.** Size exclusion chromatography with both PRPP and 9-deazaguanine.

DOI: https://doi.org/10.7554/eLife.47534.023

**Figure supplement 5.** Effect of crosslinked tetramer on activity and PRPP competition with pppGpp.

DOI: https://doi.org/10.7554/eLife.47534.025

**Figure supplement 6.** Isothermal titration calorimetry between PRPP and *B. anthracis* Hpt-1.

DOI: https://doi.org/10.7554/eLife.47534.026

**Figure supplement 7.** Molecular weight standards for size-exclusion chromatography.

DOI: https://doi.org/10.7554/eLife.47534.024

flanking dimer interface components β3 and α3 toward the active site (*Figure 4A*). While crystal contacts likely prevent loop II from completely moving over the active site (*Figure 4—figure supplement 1*), PISA analysis predicts that these changes abolish the dimer-dimer interaction. Using size-exclusion chromatography, we confirmed that apo *B. subtilis* HPRT is a tetramer (*Figure 4B*). In contrast, when PRPP, the first substrate to bind HPRT (*Yuan et al., 1992*), is added at a high concentration (500 μM) in the mobile phase, HPRT tetramers dissociate to dimers (*Figure 4B* and *Figure 4—figure supplements 2*, *3* and *4*).

We next interrogated the effect of (p)ppGpp on the oligomeric state of HPRT using the protein crosslinker dimethyl adipimidate (DMA). Crosslinked apo HPRT was resolved by SDS–PAGE as bands corresponding to monomers, dimers, trimers, or tetramers (*Figure 4C*, lane 2). All but the tetramer is likely formed by incomplete crosslinking, since dynamic light scattering showed apo *B. anthracis* Hpt-1 to be a homogeneous population with a hydrodynamic radius ($R_H$) consistent with a tetramer in solution (*Table 3*). HPRT with pppGpp remained a tetramer according to both crosslinking and dynamic light scattering experiments (*Figure 4C* and *Table 3*). Incubation of HPRT with PRPP resulted in loss of trimer and tetramer bands in SDS–PAGE (*Figure 4C*, lane 3) and a corresponding shift in tetramer to homogeneous dimer with dynamic light scattering (*Table 3*), confirming that PRPP-bound HPRT is a dimer.

Importantly, with both PRPP and pppGpp present, HPRT was tetrameric (*Figure 4C*, lanes 6 and 7 and *Table 3*), indicating that pppGpp prevents PRPP-induced dimer–dimer dissociation. Thus, pppGpp appears to selectively stabilize HPRT tetramers against PRPP-induced dissociation.

These data also suggest that preventing HPRT from dimer–dimer dissociation and keeping it as a tetramer will disfavor PRPP binding. Indeed, a cysteine substitution across the dimer–dimer interface (Y117C) forcing HPRT to stay as a tetramer via disulfide crosslink results in lower enzymatic activity, and PRPP cannot outcompete pppGpp for binding the tetrameric HPRT variant (*Figure 4—figure supplement 5*).

Taken together, these data reveal that HPRT binds PRPP as catalytically active dimers, whereas (p)ppGpp maintains HPRT as tetramers.

**Table 3.** Dynamic light scattering of *B. anthracis* Hpt-1 with pppGpp and PRPP.

| Sample | $R_H$[a] (nm) | Polyd[b] | % Polyd | Est. MW (kDa) |
|---|---|---|---|---|
| HPRT | 4.1 | 0.9 | 23.1 | 90 |
| HPRT + PRPP | 3.1 | 0.7 | 22.5 | 47 |
| HPRT + pppGpp | 4.1 | 1.0 | 24.9 | 91 |
| HPRT + both | 4.2 | 1.2 | 27.8 | 97 |

PRPP = phosphoribosyl pyrophosphate.

[a]$R_H$ = hydrodynamic radius.

[b]Polyd = polydispersity.

DOI: https://doi.org/10.7554/eLife.47534.027

## Dimer–dimer interaction allosterically positions loop II for potentiated (p)ppGpp binding

Given HPRT's oligomerization, we performed ITC experiments with pppGpp (*Figure 2—figure supplement 1*) and PRPP (*Figure 4—figure supplement 6*) to examine ligand binding cooperativity. In the conditions we tested, within physiological ranges of pppGpp and PRPP, we did not observe significant cooperativity. This is expected as cooperativity has not been observed in native bacterial and human HPRTs and has been reported only in mutant human HPRTs (*Balendiran et al., 1999*; *Guddat et al., 2002*; *Lightfoot et al., 1994*; *Patta et al., 2015*).

To understand the importance of HPRT's differential oligomeric state with substrates and (p)ppGpp, we performed interface mutational analyses, and we found that, unexpectedly, the dimer–dimer interaction is critical for HPRT's regulation by (p)ppGpp. We disrupted the dimer–dimer interface of *B. subtilis* HPRT by constructing two loop II deletion variants of different lengths that resulted in different apo oligomeric states (*Figure 4D*). While the tetrameric Δloop II (70-76) bound pppGpp as tightly as wild-type HPRT, the dimeric Δloop II (69-77) displayed strongly ablated binding to pppGpp ($K_d$ too weak to estimate; *Figure 4E*).

Since an engineered dimeric HPRT variant has weakened binding to (p)ppGpp, we next examined the oligomeric state of naturally occurring HPRT homologs with weakened inhibition by (p)ppGpp (*Figure 2*). Strikingly, in nearly all cases, the (p)ppGpp-insensitive HPRTs were also constitutive dimers (*Figure 5A*). These homologs have normal enzymatic activity (*Table 4*) and only differ in their ability to be regulated by (p)ppGpp. Our results suggest that tetrameric HPRT, but not dimeric HPRT, allows (p)ppGpp at basal levels to bind to and inhibit its activity.

The structural basis of how the dimer–dimer interaction promotes (p)ppGpp binding in tetrameric HPRTs can be seen from comparative structures of apo *B. anthracis* Hpt-1 and apo *L. pneumophila* HPRT (PDB ID 5ESW) (*Zhang et al., 2016*). Loop II in dimeric *L. pneumophila* HPRT is positioned closer to the active site (*Figure 5B*), mimicking substrates-bound HPRT even in the absence of substrates (*Figure 5—figure supplement 1*), and compresses the cavity surrounding the 5′ phosphates of ppGpp (*Figure 5C*). This conformation does not fully accommodate (p)ppGpp, but it should accommodate PRPP since its 5′ monophosphate fits in the cavity compressed by loop II (*Figure 5—figure supplement 1*). In contrast, in tetrameric HPRTs, loop II is pulled away from the active site by the dimer–dimer interaction and is positioned for optimal (p)ppGpp binding. There are many examples of the interdependence of protein oligomerization and ligand binding (*Traut, 1994*). By discovering and comparing two different evolved oligomeric states of HPRT, we have been able to demonstrate how a structural rearrangement through oligomerization allosterically affects ligand binding to a non-interface pocket.

To identify the determinant for the oligomeric state of naturally occurring HPRTs, we constructed a chimera of *B. subtilis* HPRT with 21 dimer–dimer interface residues replaced with their corresponding *L. pneumophila* HPRT residues (*Figure 5—figure supplement 2*). The chimera is a stable dimer (*Figure 5D* and *Figure 5—figure supplement 3*), indicating that the determinants for oligomerization lie in the dimer-dimer interface residues. The chimera also had a > 20 fold lower affinity for pppGpp than the tetramer ($K_d ≈ 24$ μM versus $≈ 1$ μM) (*Figure 5E*), and PRPP more potently competed with pppGpp binding to the dimeric chimera than to tetrameric WT (*Figure 5F*), suggesting that there may be an evolved linkage between the dimer–dimer interface of HPRT tetramers and (p)ppGpp regulation.

## Dimer-dimer interface coevolved with strong (p)ppGpp binding across species

The relationship between HPRT oligomeric state and sensitivity to (p)ppGpp prompted us to examine whether HPRT oligomerization has coevolved with (p)ppGpp regulation. To test this, we turned to ancestral protein sequence reconstruction to infer the evolution of HPRT (*Hochberg and Thornton, 2017*). With a phylogenetic tree derived from an alignment of 430 bacterial and eukaryotic HPRTs, we used maximum likelihood to infer the most likely ancestral HPRT protein sequences based on the phylogeny of the extant HPRT sequences (*Yang et al., 1995*) (*Figure 6A*). From the first ancestral HPRT (Anc1), the phylogenetic tree bifurcated into two broad lineages. One lineage contained the known dimeric HPRTs with weakened (p)ppGpp inhibition (*Figure 6A*) (*Hug et al., 2016*). The second lineage contained the vast majority of HPRTs, including (p)ppGpp-sensitive

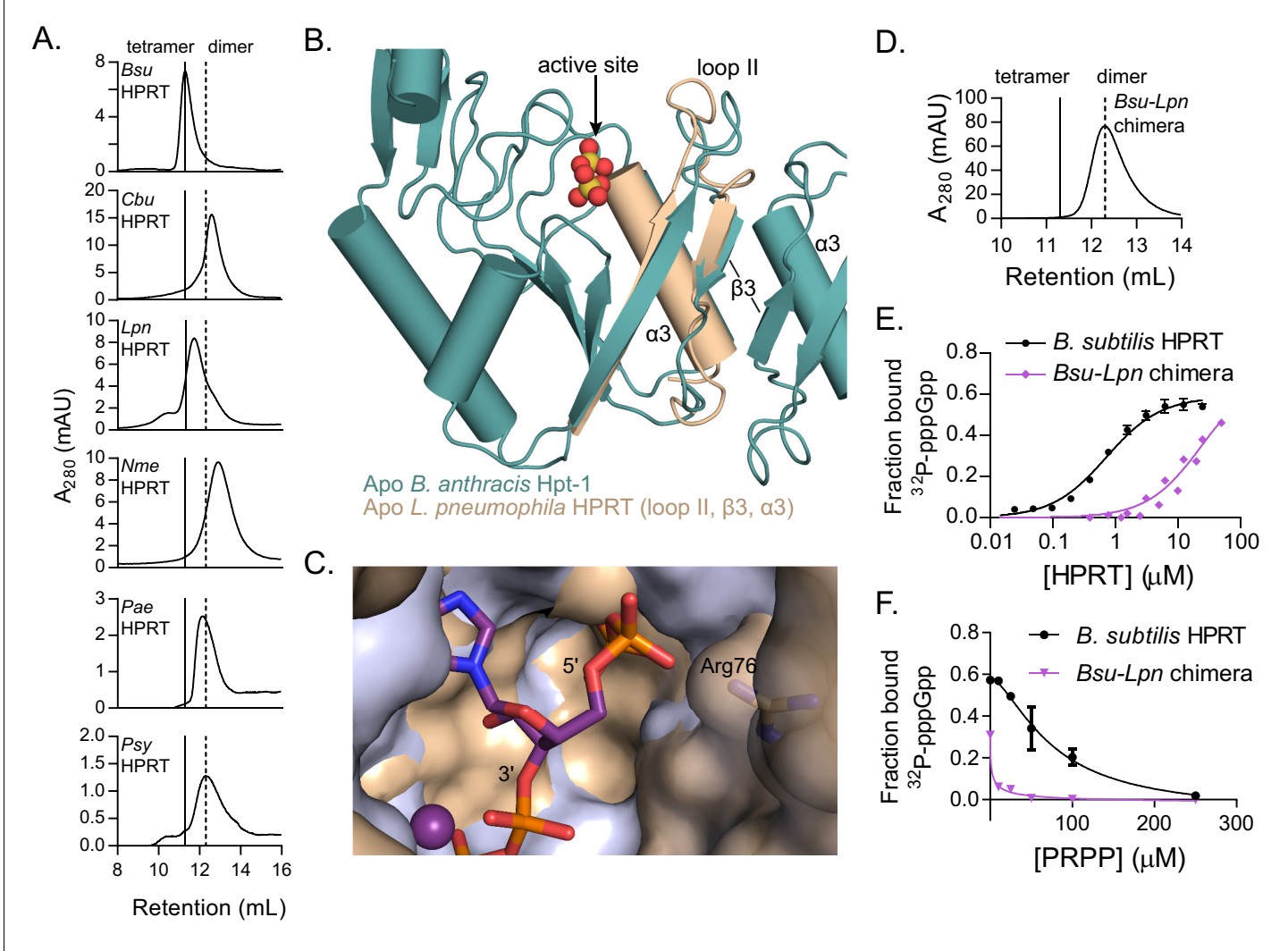

**Figure 5.** Dimer–dimer interaction holds loop II away from the (p)ppGpp binding pocket to potentiate (p)ppGpp binding. (A) Size-exclusion chromatograms show that *C. burnetii*, *L. pneumophila*, *N. meningitidis*, *P. aeruginosa*, and *P. syringae* HPRTs are non-tetrameric without ligands. Vertical lines represent *B. subtilis* HPRT tetramer and dimer peaks. (B) Overlay of apo *B. anthracis* Hpt-1 (teal) and the *L. pneumophila* HPRT interface (wheat; PDB ID 5ESW). In *L. pneumophila* HPRT, loop II and β3 are further from the dimer-dimer interface and closer to the active site. See *Figure 5—figure supplement 1* for overlay with substrates-bound HPRT. (C) Overlay of *B. anthracis* Hpt-1 – ppGpp (silver; ppGpp in purple) and *L. pneumophila* HPRT (wheat; PDB ID 5ESX) shows that the 5′ phosphate binding pocket is compressed by the conformation of loop II in the dimeric HPRT. Arg76 (side chain shown as sticks) in *L. pneumophila* HPRT forms part of the compressed pocket. Asp109 in *L. pneumophila* HPRT is hidden due to poor electron density. (D) Size-exclusion chromatogram of the dimeric *Bsu-Lpn* chimera. The chimera has 21 residues at the *B. subtilis* HPRT interface replaced with *L. pneumophila* HPRT residues (see *Figure 5—figure supplement 2*). (E) DRaCALA shows that the *Bsu-Lpn* chimera binds $^{32}$P-labeled pppGpp more weakly. Error bars represent SEM of three replicates. *Bsu-Lpn* chimera performed in duplicate. *Figure 5—figure supplement 3* shows that stability of dimeric HPRTs is not compromised. (F) Competition between PRPP and $^{32}$P-labeled pppGpp shows that the dimeric *Bsu-Lpn* chimera is more sensitive to PRPP than the tetrameric wild type. Error bars are range of duplicate.

DOI: https://doi.org/10.7554/eLife.47534.028

The following figure supplements are available for figure 5:

**Figure supplement 1.** Overlay of *L. pneumophila* HPRT and substrates-bound *B. anthracis* Hpt-1.
DOI: https://doi.org/10.7554/eLife.47534.029

**Figure supplement 2.** Residues replaces in *B. subtilis* HPRT for the *Bsu-Lpn* chimera.
DOI: https://doi.org/10.7554/eLife.47534.030

**Figure supplement 3.** Stability of dimeric HPRTs is not compromised.
DOI: https://doi.org/10.7554/eLife.47534.031

**Table 4.** HPRT kinetic parameters.

| HPRT | $K_{m\ (guanine)}$ (µM) | $K_{m\ (PRPP)}$ (µM) | $k_{cat}$ (s$^{-1}$) |
|---|---|---|---|
| *B. anthracis* 1 | 5.2 ± 1.8 | 111.6 ± 16.8 | 22 |
| *B. subtilis* | – | 165.6 ± 11.9 | 25.4 |
| *C. burnetii* | 17.6 ± 2.0 | 44 ± 3.3 | 12.9 |
| *P. phosphoreum* | – | 81 ± 12.5[a] | 30.4[a] |
| *N. meningitidis* | 68 ± 7.0 | 103 ± 12 | 9.6 |
| *E. coli*[b] | – | 192 ± 7.0[a] | 59[a] |
| *L. pneumophila*[c] | 10.5 ± 1.1 | 60 ± 9 | – |
| *M. tuberculosis*[d] | 10 ± 1 | 650 ± 70 | 0.193 |

PRPP = phosphoribosyl pyrophosphate.

[a]obtained with hypoxanthine as the purine base.

[b]**Guddat et al. (2002)**.

[c]**Zhang et al. (2016)**.

[d]**Patta et al. (2015)**.

± standard error.

DOI: https://doi.org/10.7554/eLife.47534.032

HPRTs in the Bacteroidetes, Firmicutes, and γ-proteobacteria. There are three interface residues conserved only in the dimeric lineage (Trp82, Pro86, and Ala110) (*Figure 6B*). Varying these residues in *B. subtilis* HPRT produces non-tetrameric variants (*Figure 6—figure supplement 1*). Notably, Anc1 HPRT was likely tetrameric, since it lacks the residues found in the dimeric lineage (*Figure 6B*).

A tracing of the dimer-dimer interface pinpointed residues that coevolve with (p)ppGpp regulation. We followed the change in interface residues from Anc1 to Anc2, the ancestor common to the (p)ppGpp-inhibited HPRTs (*Figure 6A*). One residue in particular, Lys81, increased in confidence between Anc1 and Anc2 (*Figure 6B*). Lys81 is part of a conserved β-strand and loop motif (residues 81–87) that interact with one another across the dimer–dimer interface, so their evolution is likely important for HPRT tetramerization (*Figure 6—figure supplement 2*). Lys81 is highly conserved in (p)ppGpp-regulated HPRTs but not in (p)ppGpp-insensitive HPRTs (*Figure 6B*), suggesting that (p)ppGpp regulation is associated with the evolution of this residue. Therefore, we constructed Lys81 variants with weakened dimer–dimer interfaces (*Figure 6C*). One variant with a charge reversal (K81E) resulted in a mostly dimeric HPRT and a less disruptive K81A HPRT exhibited a rapid equilibrium between tetramer and dimer (*Figure 6C*). Importantly, pppGpp bound less well to both K81A and K81E HPRT variants relative to wild-type HPRT (*Figure 6D*), and PRPP more potently competed with pppGpp binding to the K81A variant relative to the tetrameric wild type (*Figure 6E*). We conclude that the HPRT dimer-dimer interface has coevolved with strong (p)ppGpp regulation by sequestering loop II at the dimer–dimer interface and opening the active site for (p)ppGpp binding (*Figure 7*), whereas evolution of interface residues associated with losing this dimer-dimer interaction weakens (p)ppGpp regulation.

## Protein–protein interface allosterically alters the binding of (p)ppGpp to guanylate kinase

Intrigued by finding the determinants of (p)ppGpp regulation of HPRT, we turned to guanylate kinase (GMK), another (p)ppGpp-regulated enzyme (*Kriel et al., 2012*; *Liu et al., 2015b*). We previously found that GMK is inhibited by (p)ppGpp in multiple phyla of bacteria but not in Proteobacteria despite the fact (p)ppGpp binds the conserved active site (*Liu et al., 2015b*). This puzzle may now be explained by a protein–protein interaction affecting (p)ppGpp binding: GMK is a dimer. In the (p)ppGpp-sensitive GMK, a lid domain opens the active site for (p)ppGpp binding through pulling of the lid by the C-terminal helix of the adjoining monomer in a GMK dimer (*Figure 8A*). In the (p)ppGpp-insensitive *E. coli* GMK, the lid domain is closed, with the C-terminal helix from the adjoining monomer perhaps responsible for stabilizing the closed position (*Figure 8A*). We had previously swapped the lid domains to test their role in (p)ppGpp inhibition, but this had led to inactive

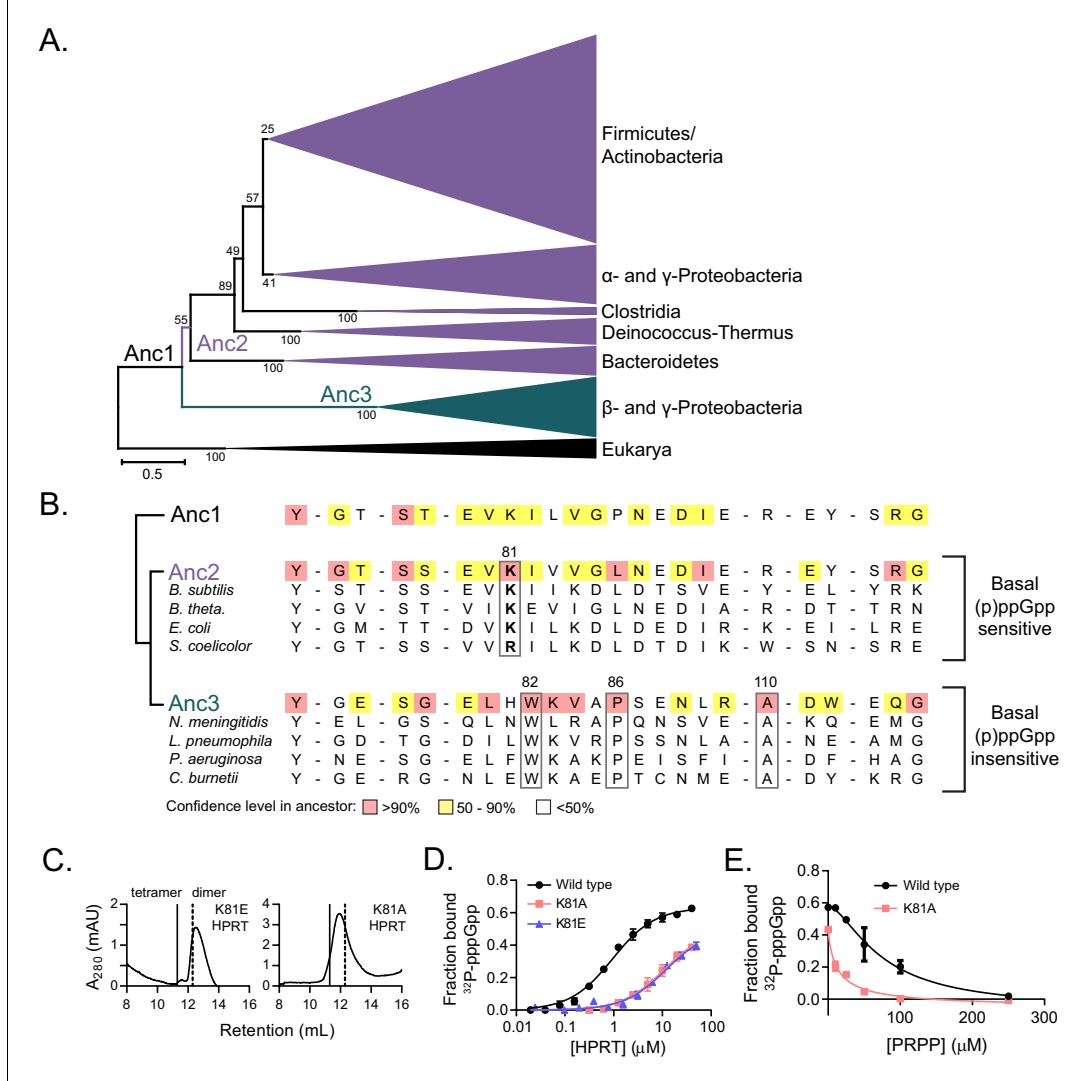

**Figure 6.** Coevolution of the HPRT dimer-dimer interface and regulation by basal levels of (p)ppGpp. (**A**) A maximum likelihood phylogenetic tree of 430 HPRT amino acid sequences rooted on eukaryotic HPRTs. Bootstrap values from 100 replicates are shown. Anc1-3 refer to ancestral HPRTs. The tree reveals two main lineages: one with dimer-dimer interaction motifs (Anc2; purple) and one lacking the interaction motifs (Anc3; blue). Branch length scale indicates 0.5 substitutions per site. See Materials and methods for tree construction. See Source Data one for the full alignment of HPRTs. (**B**) Alignment of HPRT dimer-dimer interface from Anc1, Anc2, Anc3, and select extant HPRTs. Nearly all (p)ppGpp-regulated HPRTs share Lys81 found in Anc2. Dimeric HPRTs insensitive to basal (p)ppGpp lack Lys81, but share Trp82, Pro86, and Ala110 (*B. anthracis* Hpt-1 numbering). *Figure 6—figure supplement 1* shows the role of these residues at the interface. For the ancestors, red indicates high (>90%), yellow moderate (50–90%), and no color low (<50%) confidence level in the residue identity. (**C**) *B. subtilis* K81E and K81A HPRTs have weakened tetramerization according to size-exclusion chromatography. *Figure 6—figure supplement 2* shows Lys81 at the dimer–dimer interface. (**D**) Binding of $^{32}$P-labeled pppGpp to wild type (black circles), K81A (red squares), and K81E (blue triangles) HPRTs using DRaCALA. Error bars represent SEM of three replicates. (**E**) PRPP competition with $^{32}$P-labeled pppGpp for binding wild type (black circles) and K81A HPRTs (red squares). K81A is more sensitive to PRPP competition than the tetrameric wild type. Error bars are range of duplicate.

DOI: https://doi.org/10.7554/eLife.47534.033

The following source data and figure supplements are available for figure 6:

**Source data 1.** Protein alignment of HPRTs.
DOI: https://doi.org/10.7554/eLife.47534.036
**Figure supplement 1.** Role of residues conserved in the dimer–dimer interface of dimeric HPRTs.
DOI: https://doi.org/10.7554/eLife.47534.034
**Figure supplement 2.** The conserved Lys81 bridges the dimer–dimer interface of tetrameric HPRTs.
DOI: https://doi.org/10.7554/eLife.47534.035

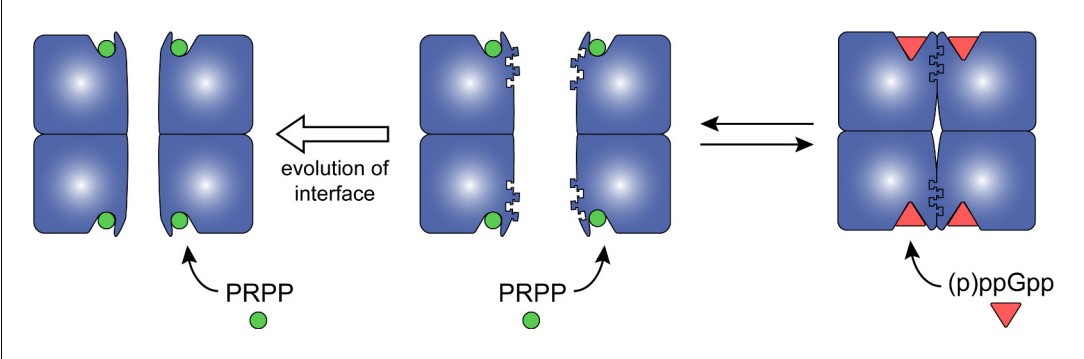

**Figure 7.** Model for how evolution of protein oligomerization affects ligand-mediated regulation. For an enzyme such as HPRT, where the inhibitor (p)ppGpp binds to nearly identical sites as the substrates (PRPP), evolutionary plasticity of inhibition can be mediated through subunit oligomerization. Dimeric HPRTs present a smaller binding pocket that preferentially bind to the substrate than to (p)ppGpp. On the other hand, HPRTs with a dimer–dimer interface motif would allosterically open the pocket, thus favoring (p)ppGpp binding and inhibition than substrate-binding and activity.
DOI: https://doi.org/10.7554/eLife.47534.039

enzymes (*Liu et al., 2015b*). To identify whether the C-terminal tail is holding the lid domain open or closed, we deleted 10 residues from the C-terminus in both the (p)ppGpp-sensitive *B. subtilis* GMK and the (p)ppGpp-resistant *E. coli* GMK with striking observations (*Figure 8B C*). Remarkably, while these residues are away from the (p)ppGpp binding pocket of the adjoining monomer, their deletion led to weakened binding between *B. subtilis* GMK and pppGpp and reduced inhibitory effect on its enzymatic activity. On the other hand, deleting the C-terminus of the (p)ppGpp-resistant *E. coli* GMK allowed a weak binding between *E. coli* GMK and pppGpp, and rendered *E. coli* GMK sensitive to pppGpp inhibition. This demonstrates that the allosteric interaction between the lid domain and C-terminus of two adjoining GMK monomers is the determining factor for (p)ppGpp specificity.

We constructed a phylogenetic tree of 41 GMKs (*Figure 8D*). As with HPRT, the tree showed that the (p)ppGpp-insensitive and (p)ppGpp-sensitive GMKs bifurcated from the last common ancestor. Importantly, these (p)ppGpp-insensitive and (p)ppGpp-sensitive GMKs use different residues to form the interaction between the lid domain and the C-terminal helix (*Figure 8E*). (p)ppGpp-insensitive GMKs have a hydrophobic interaction between the lid domain and C-terminal tail, whereas (p)ppGpp-sensitive GMKs contain more charged residues at this interface. Ancestral protein reconstruction suggests that the last common ancestor (Anc1) is more similar to the (p)ppGpp-sensitive GMKs (Anc3) (*Figure 8E*). Altogether, this suggests that the protein-protein interface of GMK has also evolved to affect (p)ppGpp binding.

## Discussion

(p)ppGpp is a stress-induced signaling molecule in bacteria that is also critical for cellular fitness and homeostasis even at basal levels. However, (p)ppGpp targets and their evolution in different bacteria remain poorly understood. Here, we have described a mechanism explaining how basal levels of (p)ppGpp potently regulate the activity of the housekeeping enzyme HPRT through a novel binding site that may represent a new class of (p)ppGpp effectors. Intriguingly, this site overlaps completely with the active site and is conserved among bacteria, yet differential regulation by (p)ppGpp can be achieved through variation of an allosteric component: the interaction between dimeric subunits of the HPRT tetramer. This interaction tethers a flexible loop at the interface and away from the active site, allowing the binding pocket to accomodate (p)ppGpp. Lack of the dimer–dimer interaction causes HPRT to occlude (p)ppGpp and favor substrate binding. This dimer–dimer interaction is due to an interface motif that appears to have co-evolved with (p)ppGpp binding in the majority of bacterial HPRTs sensitive to (p)ppGpp, whereas dimeric HPRTs without the motif are resistant to (p)ppGpp. We conclude that evolution of the dimer-dimer interface in tetrameric HPRTs has potentiated (p)ppGpp binding to enable basal (p)ppGpp modulation of metabolism, thus increasing bacterial fitness in fluctuating environments.

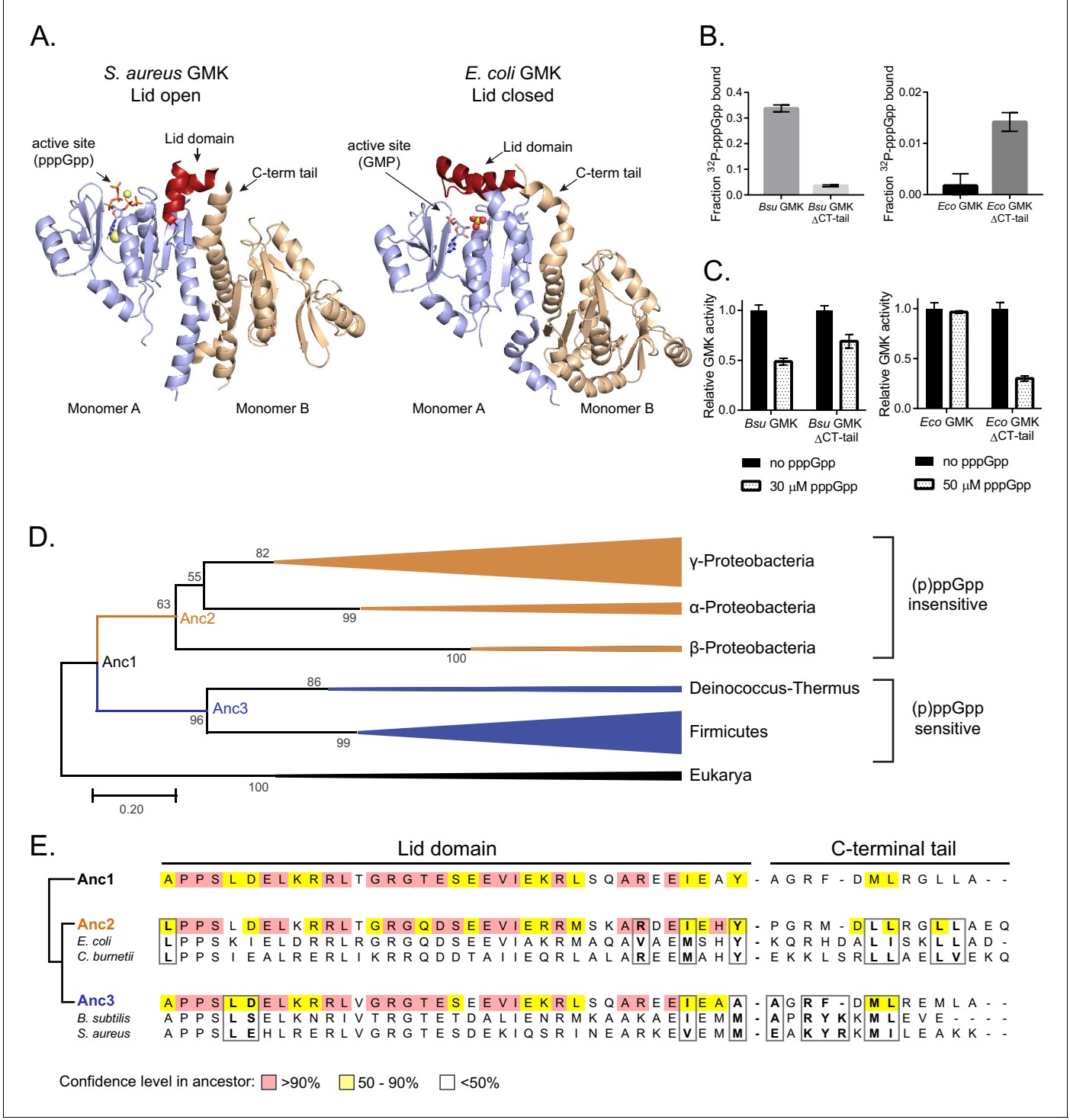

**Figure 8.** Protein–protein interface allosterically alters the binding of (p)ppGpp to guanylate kinase. (**A**) In *S. aureus* GMK (left; PDB ID 4QRH), an open lid domain (red) allows (p)ppGpp binding at the active site, but in *E. coli* GMK (right; PDB ID 2ANB), the closed lid domain occludes (p)ppGpp binding at the active site (bound to GMP). A C-terminal tail in the second monomer of a GMK dimer interacts with the lid domain in both proteins. Nineteen residues in the lid domain and six residues in the C-terminal tail of *S. aureus* GMK are unresolved. (**B**) DRaCALA of full-length and ΔC-terminal tail (ΔCT tail) *B. subtilis* and *E. coli* GMKs. Error bars are mean of duplicate. (**C**) Inhibition of GMKs from (**D**) relative to uninhibited activity. Error bars are SEM of triplicate. (**D**) Maximum likelihood phylogenetic tree of 41 GMKs rooted on eukaryotic GMKs as the outgroup. Bootstrap values from 1000 replicates are shown. Anc1-3 refer to ancestral HPRTs. See Materials and methods for tree construction. See Source Data one for the full alignment of GMKs. (**E**) Alignment of the lid domain and C-terminal tail of ancestral and select extant GMKs. Boxed residues are part of the lid domain - C-terminal tail interaction.

*Figure 8 continued on next page*

*Figure 8 continued*

DOI: https://doi.org/10.7554/eLife.47534.037

The following source data is available for figure 8:

**Source data 1.** Protein alignment of GMKs.

DOI: https://doi.org/10.7554/eLife.47534.038

## A novel, high-affinity (p)ppGpp binding motif that shares the PRPP motif

Our HPRT-ppGpp structure revealed a binding motif distinct from known (p)ppGpp-protein interactions. There are over 50 known (p)ppGpp targets across bacteria, but identifying motifs associated with (p)ppGpp binding has been difficult (*Corrigan et al., 2016*; *Wang et al., 2019*; *Zhang et al., 2018*). In a few cases, (p)ppGpp binds allosteric sites at protein interfaces (*Kanjee et al., 2011*; *Ross et al., 2013*; *Ross et al., 2016*; *Steinchen et al., 2015*; *Wang et al., 2019*). For many other targets, (p)ppGpp binds at a GTP binding site, leading to overlapping (p)ppGpp and GTP binding motifs (*Fan et al., 2015*; *Kihira et al., 2012*; *Liu et al., 2015b*; *Pausch et al., 2018*; *Rymer et al., 2012*). In the case of HPRT, however, the (p)ppGpp binding site is not at an interface between two proteins, nor is it a GTP binding site. Instead, it shares a well-characterized motif associated with PRPP binding (EDIIDSGLT in *B. anthracis* Hpt-1) (*Sinha and Smith, 2001*). Other PRPP-binding proteins, including UPRT and APRT, have been shown to bind (p)ppGpp (*Wang et al., 2019*; *Zhang et al., 2018*), and it is likely that (p)ppGpp also binds to their PRPP motif. Identification of this motif may provide a new class of (p)ppGpp-binding proteins, allowing us to predict additional targets.

(p)ppGpp interacts with proteins in two main conformations: elongated, with the phosphate arms extended away from one another in a T shape, and ring-like, with the phosphate arms near one another in a Y shape. The compact, ring-like conformation has been associated with higher affinity interactions than the elongated conformation (*Steinchen and Bange, 2016*). However, in the HPRT–ppGpp interaction, ppGpp takes an elongated conformation, but exhibits strikingly tight affinities as high as $K_d \approx 0.1$ μM for some species (*Figure 2D* and *Table 1*). This high affinity may be due to extensive backbone amide interactions with both phosphate arms as well as (p)ppGpp's close mimicry of substrate binding (*Figure 3D*). It is likely that higher affinity interactions with the elongated (p)ppGpp conformation will become more common as additional targets are characterized.

## HPRT tetramerization enables basal (p)ppGpp inhibition

Our data show that the dimer–dimer interface of HPRT potentiates (p)ppGpp regulation by allosterically promoting a conformation conducive to ligand binding (*Figure 7*). HPRT's tetramerization harnesses the flexible loop II to open the binding pocket (*Figure 4A*) for (p)ppGpp to bind with high affinity. Loop II's influence on (p)ppGpp binding may be the reason that it has evolved to be part of the interface of bacterial HPRTs, whereas in eukaryotic (e.g. human) HPRTs, which likely do not interact with (p)ppGpp in nature, loop II is on the outside of the oligomer facing solvent (*Eng et al., 2015*).

Our model can also explain the evolutionary significance of bacterial HPRTs functioning as dimers without dissociating to monomers. Strength of protein–protein interactions has been correlated with increased buried surface area at the interface (*Nooren and Thornton, 2003*). In a hypothetical monomer–monomer interaction, the smaller loop II interface may be too weak and too transient to keep loop II away from the active site. In the interaction between two homodimers, the loop II interface is duplicated, which increases the surface area and provides anchor points to hold loop II at the interface and away from the (p)ppGpp binding pocket (*Figure 7*).

HPRT tetramerization also affects a key component of basal (p)ppGpp's ability to regulate HPRT: its competition with PRPP. Most bacterial HPRTs have a $K_m$ for PRPP in the range of 50–200 μM (*Guddat et al., 2002*; *Zhang et al., 2016*) (*Table 4*). Tetrameric HPRTs that bind (p)ppGpp with high affinity also tend to have higher $K_m$ values for PRPP, suggesting that they are optimized for (p)ppGpp binding rather than PRPP binding (*Table 4*). By forcing HPRT into a dimeric state independent of ligands, either through natural evolution in the case of the HPRT homologs or through mutating the dimer–dimer interface sequence, HPRT becomes both refractory to (p)ppGpp binding

(*Figure 5*) and also preferably binds PRPP (*Figures 5F* and *6E*, *Table 4*). The overall effect is an HPRT that is resistant to inhibition by basal levels of (p)ppGpp.

## HPRT regulation allows maintenance of cellular homeostasis by basal levels of (p)ppGpp

Our characterization of HPRT explains how bacteria can be regulated by basal levels of (p)ppGpp. While (p)ppGpp is mostly chacterized as a regulator of gene expression that functions in starvation-induced concentrations, basal concentrations of (p)ppGpp are known to be responsible for sustaining antibiotic tolerance and virulence in *E. faecalis* (*Gaca et al., 2013*), maintaining cyanobacterial light/dark cycles (*Puszynska and O'Shea, 2017*), influencing rRNA expression and growth rate in *E. coli* (*Potrykus et al., 2011*), and regulating GTP synthesis in Firmicutes (*Gaca et al., 2013*; *Kriel et al., 2012*). However, the principles that determine how basal and induced (p)ppGpp regulate different cellular targets remained unclear. In the single species *B. subtilis*, for example, (p) ppGpp interacts with DNA primase ($K_{i, ppGpp} = 250$ µM), IMP dehydrogenase ($K_{i, ppGpp} = 50$ µM), and guanylate kinase ($K_{i, pppGpp} = 14$ µM), allowing them to be inhibited in vivo at induced (p)ppGpp levels (*Liu et al., 2015b*; *Pao and Dyes, 1981*; *Wang et al., 2007*). In contrast, (p)ppGpp's interaction with *B. subtilis* HPRT is far stronger ($K_i = 1.7$ µM; *Figure 1D*), potentially allowing (p)ppGpp to modulate its activity at basal levels throughout growth. Our data suggest that the dimer-dimer interaction is a mechanism that strengthens (p)ppGpp binding affinity, but also favors the inhibitor over the substrate PRPP. Physiological PRPP levels in bacteria are roughly 0.1–1 mM (*Bennett et al., 2009*; *Berlin and Stadtman, 1966*; *Jensen, 1983*). At 1 mM PRPP, the high end of physiological levels, (p)ppGpp at a basal level (25 µM) can still strongly inhibit tetrameric HPRT activity but not the constitutive dimeric homologs (*Figure 2E*). Thus tetramerization allows basal (p)ppGpp to regulate HPRT and thus constantly regulate purine nucleotide synthesis via the salvage pathway.

The basal regulation of HPRT may be important for fitness in environmental niche. For example, for bacteria in the microbiota whose HPRTs were potently inhibited by (p)ppGpp, it may robustly maintain intracellular metabolism despite fluctuations in exogenous purines that could depend on the purine content of the diet (*Choi et al., 2004*; *Kaneko et al., 2014*; *Zgaga et al., 2012*).

## Protein oligomerization allosterically alters ligand specificity

The mechanism we characterized for HPRT and GMK may represent a more broadly applicable principle by which evolution of protein oligomerization changes the conformation of a ligand binding pocket to alter ligand specificity. Oligomerization of proteins into homomers can provide multiple adaptive advantages, including mechanisms of allosteric regulation by a ligand binding to a site other than the enzyme's active, enabling cooperative substrate binding, promoting protein stability, providing complete active sites or ligand binding sites at oligomeric interfaces, maintaining proximity of signal transduction, and serving cytoskeletal or other structural roles (*Ali and Imperiali, 2005*; *Bergendahl and Marsh, 2017*; *Matthews and Sunde, 2012*; *Perutz, 1989*; *Perica et al., 2012*; *Traut, 2008*; *Traut, 1994*). For HPRT, instead, the advantage of tetramerization is strengthening (p) ppGpp specificity at the active site for enhanced competitive regulation by basal levels of this nucleotide.

Similarly, we show oligomeric interaction also plays a major role in altering specificity for the (p) ppGpp target guanylate kinase (GMK). GMK is inhibited by (p)ppGpp in multiple phyla of bacteria but not in Proteobacteria despite the fact (p)ppGpp binds the conserved active site (*Liu et al., 2015b*). We now found that this is because GMK is a dimer and the monomer–monomer interaction in GMK allosterically affects the lid domain conformation and (p)ppGpp binding by occluding (p) ppGpp binding in *E. coli* and promoting (p)ppGpp binding in *B. subtilis*. By deleting the C-terminal helix from the adjoining monomer that is responsible for stabilizing the lid domain position, we can, for the first time, remove the differential specificity to (p)ppGpp between *E. coli* and *B. subtilis* GMK homologs while keeping the enzymes active (*Figure 8*).

It is striking to note that a majority of ligands bind within 6 Å of a protein–protein interface (*Gao and Skolnick, 2012*). Thus, the allosteric effect of oligomeric or other protein-protein interactions on ligand binding likely extends to many other regulations by signaling ligands.

## Coevolution of ligand binding and protein oligomerization

What is the evolutionary advantage of regulating ligand-binding through protein-protein interactions? One potential advantage is that it provides evolutionary flexibility for ligand binding. Evolving different residues within ligand binding sites can alter ligand specificity, but active sites that bind substrates and inhibitors are under functional and evolutionary constraints to maintain enzymatic activity (*Echave et al., 2016*; *Huang et al., 2015*). On the other hand, protein–protein interfaces like the dimer–dimer interface of HPRT and the monomer–monomer interface of GMK are more evolutionarily flexible, particularly when they are not obligate interactions, as they allow mutations to accumulate away from a conserved active site (*Echave et al., 2016*; *Mintseris and Weng, 2005*). This suggests that changing oligomeric states could be an evolutionarily flexible mechanism for altering ligand specificity (*Figure 7*). Indeed, the coevolution of (p)ppGpp regulation and the HPRT and GMK protein–protein interactions has provided us with examples of how protein oligomerization and ligand binding can coevolve, demonstrating that organisms have already adopted this strategy.

In addition to (p)ppGpp, many small molecules serve as signaling ligands that regulate protein activities (*Gao and Skolnick, 2012*; *Najmanovich, 2017*). Our results suggest that ligand binding, even at non-interface binding pockets, influence evolutionary diversification of protein oligomers potentially through purifying selection of conformations that favor protein–ligand interactions. While homomeric evolution of some proteins has been implicated as physicochemical or stochastic processes (*Abrusán and Marsh, 2018*; *André et al., 2008*; *Lukatsky et al., 2007*; *Lynch, 2013*), our data provide evidence for ligand binding as an adaptive advantage driving the evolutionary diversification of protein homomers. Given the proximity of ligand binding sites to protein interfaces (*Gao and Skolnick, 2012*), and since it is easier to evolve protein–protein interactions (*Perica et al., 2012*) rather than evolving new sites for allosteric regulation, such an adaptive benefit is likely to exist more broadly beyond (p)ppGpp in other protein–ligand interactions.

# Materials and methods

### Key resources table

| Reagent type (species) or resource | Designation | Source or reference | Identifiers | Additional information |
|---|---|---|---|---|
| Strain, strain background (*Bacillus subtilis*) | gmk::ecgmk | (*Liu et al., 2015b*) | N/A | |
| Strain, strain background (*B. subtilis*) | gmk::ecgmk amyE::Phyperspac-ecgpt | This work | N/A | |
| Strain, strain background (*B. subtilis*) | gmk::ecgmk amyE::Phyperspac-bshprT | This work | N/A | |
| Recombinant DNA reagent | pLIC-trPC-HA (plasmid) | (*Stols et al., 2002*) | N/A | |
| Recombinant DNA reagent | pHM1381 (plasmid) | (*Mechold et al., 2002*) | N/A | |
| Recombinant DNA reagent | pDR90 (plasmid) | (*González-Pastor et al., 2003*) | N/A | |
| Peptide, recombinant protein | human HPRT1 | Novoprotein | Novoprotein #C294 | |
| Chemical compound, drug | 5-phospho-D-ribose 1-diphosphate pentasodium salt (PRPP) | MilliporeSigma | MilliporeSigma # P8296 | |
| Chemical compound, drug | dimethyl adipimidate (DMA) | ThermoFisher | ThermoFisher # 5001417627 | |

*Continued on next page*

*Continued*

| Reagent type (species) or resource | Designation | Source or reference | Identifiers | Additional information |
|---|---|---|---|---|
| Chemical compound, drug | 9-deazaguanine | Santa Cruz Biotechnology | Santa Cruz #sc-217528 | |
| Chemical compound, drug | SYPRO Ruby | Bio-Rad | Bio-Rad # 1703125 | |
| Chemical compound, drug | SYPRO Orange | MilliporeSigma | MilliporeSigma #S5692 | |
| Chemical compound, drug | guanine | MilliporeSigma | MilliporeSigma #G6779 | |
| Software, algorithm | GraphPad Prism v. 5.02 (software) | https://www.graphpad.com/ | RRID:SCR_002798 | |
| Software, algorithm | ImageQuant (software) | https://www.gelifesciences.com | RRID:SCR_014246 | |
| Software, algorithm | MicroCal Origin 5.0 (software) | https://www.malvernpanalytical.com | RRID:SCR_002815 | |
| Software, algorithm | MEGA X (software) | https://www.megasoftware.net/ | RRID:SCR_000667 | |
| Software, algorithm | Phenix (software) | https://www.phenix-online.org/ | RRID:SCR_014224 | |
| Software, algorithm | Coot (software) | https://www2.mrc-lmb.cam.ac.uk/personal/pemsley/coot/ | RRID:SCR_014222 | |
| Software, algorithm | Dynamics v. 5.25.44 (software) | https://www.wyatt.com/products/software/dynamics.html | N/A | |

## Strain and plasmid construction and mutagenesis

For purification and DRaCALA analyses, *hprT* coding sequences were cloned into the pLIC-trPC-HA vector (pJW269) using the ligation independent cloning (LIC) protocol (*Eschenfeldt et al., 2009*; *Stols et al., 2002*) and was scaled up to include more *hprT* sequences when necessary as described (*Abdullah et al., 2009*). The *hprT* sequences were identified in each organism by enzyme classification number (EC 2.4.2.8), by BLAST using *B. subtilis* or *E. coli* hprTs, or by database annotation (Uni-Prot, EnsemblBacteria, KEGG Genome). Bacterial *hprT* homologs were amplified from genomic DNA, cloned using LIC, and transformed into *E. coli* BL21(DE3) (see *Supplementary file 1* for primers and plasmids). Amino acid substitutions were performed either with QuikChange XL Site-Directed Mutagenesis (Agilent Technologies) or with a megaprimer site-directed mutagenesis protocol (*Kirsch and Joly, 1998*). Loop II deletions were made using a protocol as described (*Hansson et al., 2008*). pLIC-trPC-HA inserts were amplified and sequenced for confirmation using oJW1124 and oJW492.

To create strains overexpressing protein in *B. subtilis gmk::ecgmk*, *B. subtilis hprT* and *E. coli gpt* were amplified with oJW1569/1570 and oJW1550/1551, respectively, prior to restriction cloning into pDR90 using HindIII and SphI (New England Biolabs). Plasmids were linearized with XhoI and transformed into JDW2108 (*B. subtilis gmk::ecgmk*) by growing JDW2108 in 1x modified competence medium (*Spizizen, 1958*) until turbid (≈6 hr) prior to adding linearized plasmid and growing an additional 2 hr. Transformants were selected on 80 μg/ml spectinomycin (MilliporeSigma) and recombination into *amyE* was confirmed by patching transformants on LB-starch plates and checking for lack of starch utilization.

## Protein purification

HPRTs were recombinantly expressed in *E. coli* BL21(DE3) (NEB) from a pLIC-trPC-HA plasmid with the gene inserted downstream of a sequence encoding a 6X histidine tag and a tobacco etch virus (TEV) protease recognition site. Seed cultures grown to mid-log phase were diluted 1:50 into batch culture of LB supplemented with 100 μg/mL carbenicillin (Research Products International). Protein synthesis was induced at $OD_{600} \approx 0.8$ with 1 mM isopropyl β-D-1 thiogalactopyranoside (IPTG) (RPI)

for 4 hr. Cells were pelleted and stored at −80° C until purification. GMKs and *E. coli* XGPRT were expressed in an identical manner.

For large-scale purifications, cells were resuspended in Lysis Buffer (50 mM Tris-HCl pH 7.5, 500 mM NaCl, 10 mM imidazole) and lysed with a French press. The lysate was centrifuged to obtain the soluble fraction, which was filtered through 0.45 μm filters. The sample was put over a HisTrap FF column (GE Healthcare) on an AktaPure FPLC (GE Healthcare), the column was washed with 20 column volumes of Lysis Buffer, and the protein was eluted with a gradient of increasing Elution Buffer (50 mM Tris-HCl pH 7.5, 500 mM NaCl, 250 mM imidazole). Recombinant protein fractions were dialyzed with 10 mM Tris-HCl (pH 7.5), 100 mM NaCl, 1 mM DTT, and 10% glycerol prior to concentrating and flash-freezing for storage. For small-scale purifications of HPRTs, GMKs, and XGPRT, Ni-NTA spin columns were used according to manufacturer's instructions (Qiagen). Lysis buffer was 100 mM sodium phosphate (pH 8.0), 500 mM NaCl, and 10 mM imidazole. Wash and elution buffers were the same as the lysis buffer except with 20 mM and 500 mM imidazole, respectively. Protein purity was determined using SDS-PAGE, and concentrations were measured using the Bradford assay (Bio-Rad) or using $A_{280}$ with extinction coefficients calculated by ProtParam (SIB ExPASy Bioinformatics Resource Portal).

To purify 35 HPRT homologs for activity assays (see *Figure 2*), a 96-well Capturem His-tagged purification kit (Clontech) was used according to the manufacturer's instructions. The following buffers were used: lysis [xTractor (Clontech) + 1 μg/mL DNase I], wash [20 mM $Na_3PO_4$ pH 7.6, 200 mM NaCl, 10 mM imidazole], elution [20 mM $Na_3PO_4$ pH 7.6, 500 mM NaCl, 500 mM imidazole]. Following purification, the buffer was exchanged to 10 mM HEPES pH 8, 150 mM NaCl, 10 mM $MgCl_2$, 1 mM DTT using Zeba spin desalting plates (Thermo Scientific) according to the manufacturer's instructions. The Bradford assay was used to measure protein concentration, and the proteins were aliquoted at 8 μM and flash-frozen with liquid nitrogen.

For crystallography with ppGpp, recombinant *B. anthracis* Hpt-1 was purified as described above followed by dialysis with His-tagged TEV protease in 10 mM Tris-HCl (pH 7.5), 100 mM NaCl, and 1 mM DTT. The dialyzed protein was incubated with Ni-NTA beads for 30 min, the beads were centrifuged and the supernatant was run over a HiPrep Sephacryl 16/60 S-100 HR column (GE Healthcare) in 10 mM Tris-HCl pH 8.3, 250 mM NaCl, and 1 mM DTT. Relevant fractions were concentrated to ≈10 mg/mL. For crystallography with substrates, the same protocol was followed but with 10 mM Tris-HCl pH 8 and 100 mM NaCl as the size exclusion buffer.

## Enzyme inhibition assays

HPRT activity assays were performed as described previously (*Biazus et al., 2009*; *Kriel et al., 2012*; *Xu et al., 1997*). The standard assay was performed at 25°C and contained 100 mM Tris-HCl (pH 7.4), 12 mM $MgCl_2$, 1 mM PRPP (MilliporeSigma), 50 μM guanine or hypoxanthine (MilliporeSigma), and 20 nM HPRT (measured by monomer). Reactions were initiated with the purine base and monitored in a spectrophotometer (Shimadzu UV-2401PC) at 257.5 nm or 245 nm for conversion of guanine to GMP or hypoxanthine to IMP, respectively. A difference in extinction coefficients of 5900 $M^{-1}cm^{-1}$ was used for GMP and guanine and 1900 $M^{-1}cm^{-1}$ for IMP and hypoxanthine. For inhibition curves, assays were performed at the substrate concentrations listed above and at variable pppGpp concentrations. (p)ppGpp was synthesized as described (*Liu et al., 2015b*). Initial velocities of the inhibited reactions were normalized to the uninhibited initial velocity prior to fitting to the equation $Y = 1/(1 + (x / IC_{50})^s)$ to calculate $IC_{50}$. GMK activity assays were performed as previously described (*Liu et al., 2015b*), except an enzyme concentration of 20 nM and a substrate concentration of 100 μM GMP were used. Due to lower activity of the *E. coli* ΔCT-tail variant, assays were performed at 200 nM enzyme rather than 20 nM enzyme for *E. coli* GMK.

To test inhibition of HPRT homologs (see *Figure 1*), reactions were performed in a Synergy Two microplate reader (BioTek). The assay was performed at 25°C with 50 μM hypoxanthine, 1 mM PRPP, and 100 nM HPRT (measured by monomer) in the same reaction buffer as above. For *Coxiella burnetii* HPRT, 50 μM guanine was used as the substrate since its activity was very low with hypoxanthine. Reactions were performed in triplicate without (p)ppGpp, with 25 μM ppGpp, and with 25 μM pppGpp. Reaction rates from the first-order kinetic curves were determined using R (v 3.4.3).

For kinetic studies, pppGpp concentrations were varied between 3.125 and 50 μM and PRPP concentrations were varied between 0.03125 and 2.5 mM. Data were analyzed and fitted to a global Michaelis-Menten competitive inhibition model using GraphPad Prism v5.02. PRPP provided by

MilliporeSigma was listed as $\geq$75% pure. For the purposes of kinetic experiments, the purity was assumed to be 100%.

## Isothermal titration calorimetry

Experiments were performed using the MicroCal iTC$_{200}$ (GE Healthcare). *B. anthracis* Hpt-1 was dialyzed into the ITC buffer (10 mM HEPES pH 8, 150 mM NaCl, 10 mM MgCl$_2$) with three buffer changes at 4°C. The concentration of protein was calculated using a molar extinction coefficient of 16390 M$^{-1}$cm$^{-1}$ and A$_{280(true)}$ (A$_{280}$ – (1.96 × A$_{330}$) (*Pace et al., 1995*). The experiments were performed at 25°C with 45.5 µM HPRT (measured by monomer), a reference power of 6 µCal/s, and a stirring speed of 1000 RPM. pppGpp was solubilized in dialysate from protein dialysis and its concentration was measured using a molar extinction coefficient of 13,700 M$^{-1}$cm$^{-1}$ and A$_{253}$. pppGpp (500 µM) was titrated into *B. anthracis* Hpt-1 with the following: 1 × 1 µL (discarded), 19 × 2 µL. ITC between PRPP and Hpt-1 were performed in identical conditions. PRPP was solubilized in protein dialysate at a concentration of 1 mM. For data analysis, PRPP was assumed to be at a concentration of 820 µM, since the PRPP lot analysis from MilliporeSigma listed the PRPP as 82% pure. Data analysis and one-site binding modeling were performed using MicroCal Origin 5.0 software.

## X-ray crystallography

Proteins were prepared for crystallography as described above. For crystals formed with pppGpp, *B. anthracis* Hpt-1 was concentrated to ≈10 mg/mL in 10 mM Tris pH 8.3, 250 mM NaCl, and 1 mM DTT. The pppGpp ligand was resuspended in ddH$_2$O. MgCl$_2$ was added to the protein at a final concentration of 1 mM and crystals were formed using hanging drop vapor diffusion with 900 µL of reservoir liquid in each well. Each drop contained 0.9 µL protein, 0.9 µL reservoir liquid, and 0.2 µL pppGpp (final concentration, 1.5 mM pppGpp). Crystals formed in 0.2 M ammonium tartrate dibasic pH 6.6% and 20% PEG 3350 in 3–6 months. Identical crystals formed in replicated conditions in 1–2 weeks. Crystals were soaked in reservoir liquid with 25% glycerol prior flash freezing in liquid nitrogen. Apo protein crystals formed within 1–2 days in multiple conditions with high sulfate concentrations. Using protein preparations from above, apo crystals formed in 0.01 M CoCl$_2$, 0.1 M MES monohydrate pH 6.5, 1.8 M ammonium sulfate, and crystals were soaked in reservoir liquid with 25% glycerol for cryoprotection prior to freezing.

For crystals with substrates, *B. anthracis* Hpt-1 was concentrated to ≈10 mg/mL in 10 mM Tris pH 8 and 100 mM NaCl. Additives were diluted in the protein solution at final concentrations of 10 mM MgCl$_2$, 2 mM PRPP, and 1 mM 9-deazaguanine (Santa Cruz Biotechnology) prior to crystallization. PRPP was resuspended in ddH$_2$O and 9-deazaguanine was resuspended in 100% DMSO. Drops for hanging drop vapor diffusion comprised 1 µL crystal condition and 1 µL protein/ligand mixture. Crystals formed within 3 days in 0.2 M ammonium acetate, 0.1 M sodium acetate trihydrate pH 4.6, 30% PEG 4000. Reservoir solution with 25% ethylene glycol was added to the drops for cryoprotection prior to flash freezing in liquid nitrogen.

Diffraction data was collected at the Life-Science Collaborative Access Team (LS-CAT), beamline 21-ID-F (Hpt-1 with sulfates), 21-ID-G (Hpt-1 with ppGpp), and 21-ID-D (Hpt-1 with substrates) at the Advanced Photon Source (APS) at Argonne National Labs (Argonne, IL). Data was indexed and scaled using HKL2000 (*Otwinowski and Minor, 1997*). Phasing for Hpt-1 with sulfates and ppGpp was determined by molecular replacement with PDB ID 3H83 as a search model using Phenix (*Adams et al., 2010*). Phasing for Hpt-1 with substrates was determined by molecular replacement with Hpt-1-ppGpp as a search model using Phenix. Iterative model building with Coot and refinement with Phenix produced final models (*Emsley and Cowtan, 2004*).

Predicted oligomeric states of crystallographic assemblies were determined by PISA ('Protein interfaces, surfaces, and assemblies' service PISA at the European Bioinformatics Institute) (*Krissinel and Henrick, 2007*).

## Size exclusion chromatography

Size exclusion chromatography was performed using AktaPure and a Superose 12 10/300 GL column (GE Healthcare, Inc). A buffer consisting of 10 mM HEPES pH 8.0, 100 mM NaCl, and 10 mM MgCl$_2$ was used to run ≈10–20 µM (0.2–0.5 mg/mL) protein over the column at 0.1–0.25 mL/min. Additional 1 mM DTT was used for proteins containing cysteines. For gel filtration with ligands, 500 µM

PRPP and 100 µM 9-deazaguanine were included in the mobile phase buffer where necessary. A gel filtration standard (Bio-Rad) was used to establish molecular weight, and bovine serum albumin (BSA) was included as an additional marker.

## Dimethyl adipimidate crosslinking

Crosslinking was performed with ≈10 µM *B. subtilis* HPRT (measured by monomer), 20 mM dimethyl adipimidate (DMA) (Thermo Scientific), and 500 µM ligand. DMA was suspended in 25 mM HEPES pH 8.0, 100 mM NaCl, 10 mM $MgCl_2$, and 10% glycerol, and the solution was buffered to pH 8.5. HPRT was dialyzed into 25 mM HEPES pH 7.5, 100 mM NaCl, and 10% glycerol. Ligands were incubated with protein for 10 min followed by a 15-min incubation with DMA at room temperature. Reactions were terminated with addition of 2X Laemmli buffer (Bio-Rad) for immediate analysis with SDS-PAGE (10% polyacrylamide gel). Gels were stained with SYPRO Ruby (Bio-Rad) according to the manufacturer's protocol and imaged using a Typhoon FLA9000 (GE Healthcare).

## Dynamic light scattering

Dynamic light scattering was performed using DynaPro99 (Protein Solutions/Wyatt Technologies). Readings were from a 20 µL solution (10 mM Tris-HCl pH 8, 100 mM NaCl, 10 mM $MgCl_2$, 6.5% glycerol) with 4 mg/mL (≈175 µM) *B. anthracis* Hpt-1 and 2 mM ligands. Data were analyzed using Dynamics version 5.25.44 software.

## DRaCALA

DRaCALA was performed with pure protein and radioactive ligand as described (*Roelofs et al., 2011*). [5' α-$^{32}$P] pppGpp was synthesized according to modified protocols of non-radioactive and radioactive pppGpp syntheses (*Corrigan et al., 2016*; *Hogg et al., 2004*; *Mechold et al., 2002*). The reaction contained 25 mM bis-Tris propane (pH 9.0), 15 mM $MgCl_2$, 0.5 mM DTT, 2 mM ATP, 2 µM Rel$_{seq}$ (1-385), and 37.5 µCi [α-$^{32}$P] GTP (Perkin Elmer). The reaction was incubated at 37°C for 1 hr. The reaction was diluted in 0.5 mL of Buffer A (0.1 mM LiCl, 0.5 mM EDTA, 25 mM Tris-HCl pH 7.5) prior to adding to a 1 mL HiTrap QFF strong anion exchange column (GE Healthcare) equilibrated with 10 column volumes (CV) of Buffer A. The column was washed with 10 CV of Buffer A followed by an additional wash with 10 CV of 83% Buffer A + 17% Buffer B (Buffer B: 1 M LiCl, 0.5 mM EDTA, 25 mM Tris-HCl pH 7.5). $^{32}$P-pppGpp was eluted with a mixture of 50% Buffer A + 50% Buffer B. Fractions of 1 mL were collected from the elution.

DRaCALA reactions (20 µL) using purified HPRT contained 100 mM Tris-HCl (pH 7.4), 12 mM $MgCl_2$, protein diluted to appropriate concentrations in 20 mM HEPES pH 8 and 100 mM NaCl, and $^{32}$P-pppGpp (1:100 final dilution of first elution fraction from $^{32}$P-pppGpp purification). DRaCALA reactions using GMK contained 100 mM Tris-HCl (pH 7.4), 100 mM KCl, and 10 mM $MgCl_2$. Reactions with *B. subtilis* GMK and ΔCT-tail were performed with 20 µM protein and reactions with *E. coli* GMK and ΔCT-tail were performed with 40 µM protein. All protein concentrations were measured by monomer. For competition experiments between $^{32}$P-labeled pppGpp and PRPP, non-radioactive PRPP was resuspended in $ddH_2O$ and added at the concentrations specified. Protein was dialyzed or diluted into buffer lacking glycerol, as glycerol interferes with diffusion of the aqueous phase. Reactions were incubated at room temperature for 10 min. Two microliters from each reaction were spotted in duplicate on Protran BA85 nitrocellulose (GE Healthcare) via pipette or a replicator pinning tool (VP 408FP6S2; V and P Scientific, Inc). Spots were allowed to dry and radioactivity was detected with phosphorimaging (Typhoon FLA9000). Fraction bound of $^{32}$P-pppGpp was calculated as described (*Roelofs et al., 2011*). Data were analyzed in GraphPad Prism v5.02 and fitted to the equation $Y = (B_{max} \times K_d) / (K_d + X)$ (*Roelofs et al., 2011*). PRPP competition curves were fitted with a four-parameter logistical (4PL) model.

DRaCALA using cell lysates was adapted from a previous protocol (*Roelofs et al., 2015*). One milliliter of cells containing overexpressed recombinant HPRT was pelleted and resuspended in 100 µL of binding buffer (20 mM Tris-HCl pH 8, 100 mM NaCl, 12 mM $MgCl_2$, 0.1 mM DTT) supplemented with 1 µM PMSF, 250 µg/mL lysozyme, and 10 µg/mL DNase I. Cells were lysed with three freeze/thaw cycles. In a 20 µL DRaCALA reaction, 10 µL of cell lysate was added to binding buffer and $^{32}$P-pppGpp. Reactions were performed and analyzed as above. For measuring binding affinity ($K_d$) of proteins in cell lysates, recombinant protein expression level in cell lysates was

determined by comparing expression to a standard of purified *B. subtilis* HPRT co-resolved with SDS-PAGE.

## Differential scanning fluorimetry

Differential scanning fluorimetry was performed using 10 µM protein (measured by monomer) in a buffer containing 20 mM HEPES pH 8, 100 mM NaCl, 10 mM MgCl$_2$, 1 mM DTT, and 10X SYPRO orange dye (diluted from 5000X stock; MilliporeSigma). Proteins were mixed in an optically clear quantitative PCR (qPCR) 96-well plate and sealed with plastic film. Relative fluorescence intensity was monitored in a Bio-Rad qPCR machine using FRET detection over a temperature increase of 1˚ C/min from 25˚C to 90˚C. Data were analyzed using GraphPad Prism.

## Thin layer chromatography

Thin layer chromatography was performed as described previously (*Bittner et al., 2014*; *Schneider et al., 2003*). Cells were grown in a low-phosphate medium with casamino acids (S7 defined medium [*Vasantha and Freese, 1980*] supplemented with 0.1% glutamate, 1% glucose, and 0.5% casamino acids; low-phosphate medium contained one-tenth the phosphate concentration as S7 defined medium). JDW2121 and JDW2128 were grown with 1 mM IPTG to induce expression of exogenous phosphoribosyltransferases. Cells were labeled with 50 µCi/ml $^{32}$P orthophosphate (900 mCi/mmol; PerkinElmer) at an OD ≈ 0.02–0.05 and grown to an OD ≈ 0.15–0.2 prior to treating with 1 mM guanosine (MilliporeSigma). Guanosine stock was 100 mM diluted in 100% DMSO. At given timepoints following guanosine addition, nucleotides were extracted by adding 100 µl of sample to 20 µl 2 N formic acid and incubating on ice for at least 20 min. Samples were centrifuged at max speed for 15 min and supernatant (≈60 µl) was transferred to a new tube. Samples were spotted on PEI cellulose plates (MilliporeSigma) and developed in 1.5 M KH$_2$PO$_4$ pH 3.4. Plates were exposed to a phosphor screen and scanned on a Typhoon scanner. Nucleotide levels were quantified by subtracting background signal in each lane and expressing the level as a ratio to the untreated sample (t = 0).

## Protein sequence analysis and ancestral sequence reconstruction

To construct a 16S rRNA tree, sequences were obtained from the Ribosomal Database Project (*Cole et al., 2014*) or from NCBI. Sequences were aligned with ClustalW in MEGA X. MEGA X was used to identify the best substitution model for evolution as the General Time Reversible model with gamma distributed substitution rates. The maximum likelihood phylogenetic tree was constructed with this model in MEGA X with 1000 bootstrap replicates. Eukaryotic 18S rRNA sequences for *H. sapiens*, *C. elegans*, and *S. cerevisiae* were used as the outgroup.

For HPRT phylogenetic analysis, protein sequences were obtained in UniProt either by searching for HPRT's enzyme classification number (EC 2.4.2.8) in a given organism or by using NCBI's and UniProt's BLAST algorithms to find HPRTs similar (>50% identity) to a model organism representing different clades (e.g. *B. subtilis*, *E. coli*, *S. aureus*, *S. coelicolor*, *B. thetaiotaomicron*, *L. pneumophila*, *P. aeruginosa*, *T. gondii*, *C. elegans*, *H. sapiens*). Protein sequences were aligned using MUSCLE in MEGA X with a gap-opening penalty of −3.2 and a gap-extending penalty of −0.2. See *Figure 6—source data 1* for the complete alignment. The most appropriate evolutionary model for amino acid substitution was identified using ProtTest 3 (*Darriba et al., 2011*) as the Le and Gascuel model (*Le and Gascuel, 2008*) with gamma distributed substitution rates (four categories) and invariant sites. The maximum likelihood phylogenetic tree was constructed using this substitution model in MEGA X with 100 bootstrap replicates and a nearest neighbor interchange topology search. The clade containing eukaryotic HPRTs was used as the outgroup. Ancestral HPRT sequences were reconstructed using a marginal probability reconstruction in MEGA X with the phylogenetic tree and the same substitution model used to obtain the tree (*Yang et al., 1995*).

For GMK phylogenetic analysis, protein sequences were obtained from UniProt by searching for EC 2.7.4.8 in the organisms studied in *Liu et al. (2015b)*. Related organisms were identified with UniProt's BLAST algorithm. Protein sequences were aligned using MUSCLE in MEGA X with default settings. See *Figure 8—source data 1* for the complete alignment. ProtTest3 identified the most appropriate substitution model as the Le and Gascuel model (*Le and Gascuel, 2008*) with gamma

distributed substitution rates (four categories) and invariant sites. The phylogenetic tree and ancestral reconstruction were performed as they were with HPRT.

## Acknowledgements

We thank Esra Gumuser and Matthew Macgilvray for technical help, and members of the Wang lab for feedback on this manuscript. We thank Federico Rey, Bob Kerby, JD Sauer, Dan Pensinger, Helen Blackwell, and Kayleigh Nyffeler for assistance in obtaining bacterial genomic DNA. We thank Katrina Forest for assistance with dynamic light scattering. This research used resources of the Advanced Photon Source, a U.S. Department of Energy (DOE) Office of Science User Facility operated for the DOE Office of Science by Argonne National Laboratory under Contract No. DE-AC02-06CH11357. Use of Life Sciences Collaborative Access Team beamline 21-ID-D was supported by the Michigan Economic Development Corporation and the Michigan Technology Tri-Corridor (Grant 085P1000817). This work was funded by NIH R35 GM127088 and R01 GM084003 to JDW, and BWA was supported by NSF GRFP DGE-1256259.

## Additional information

### Funding

| Funder | Grant reference number | Author |
|---|---|---|
| National Institute of General Medical Sciences | R35 GM127088 | Jue D Wang |
| Howard Hughes Medical Institute | Faculty Scholar | Jue D Wang |
| National Science Foundation | GRFP DGE-1256259 | Brent W Anderson |
| National Institutes of Health | R01 GM084003 | Jue D Wang |

The funders had no role in study design, data collection and interpretation, or the decision to submit the work for publication.

### Author contributions

Brent W Anderson, Conceptualization, Data curation, Formal analysis, Validation, Investigation, Visualization, Methodology, Writing—original draft, Writing—review and editing; Kuanqing Liu, Conceptualization, Formal analysis, Investigation, Methodology; Christine Wolak, Katarzyna Dubiel, Formal analysis, Investigation, Writing—review and editing; Fukang She, Investigation; Kenneth A Satyshur, Data curation, Formal analysis, Investigation; James L Keck, Resources, Data curation, Supervision, Visualization, Writing—review and editing; Jue D Wang, Conceptualization, Resources, Supervision, Funding acquisition, Project administration, Writing—review and editing

### Author ORCIDs

Brent W Anderson (iD) https://orcid.org/0000-0003-2785-5343
Kenneth A Satyshur (iD) http://orcid.org/0000-0001-9371-2493
Jue D Wang (iD) https://orcid.org/0000-0003-1503-170X

### Decision letter and Author response

Decision letter https://doi.org/10.7554/eLife.47534.055
Author response https://doi.org/10.7554/eLife.47534.056

## Additional files

### Supplementary files

• Supplementary file 1. Primers, plasmids, and strains.
DOI: https://doi.org/10.7554/eLife.47534.040

• Transparent reporting form

DOI: https://doi.org/10.7554/eLife.47534.041

## Data availability

Diffraction data have been deposited in PDB under the accession codes 6D9Q (https://www.rcsb.org/structure/6d9q), 6D9R (https://www.rcsb.org/structure/6d9r), and 6D9S (https://www.rcsb.org/structure/6D9S). All data generated or analysed during this study are included in the manuscript and supporting files. Source data files have been provided for Table 2.

The following datasets were generated:

| Author(s) | Year | Dataset title | Dataset URL | Database and Identifier |
|---|---|---|---|---|
| Satyshur KA, Dubiel K, Anderson B, Wolak C, Keck JL | 2019 | The sulfate-bound crystal structure of HPRT (hypoxanthine phosphoribosyltransferase) | https://www.rcsb.org/structure/6D9Q | Protein Data Bank, 6D9Q |
| Satyshur KA, Dubiel K, Anderson B, Wolak C, Keck JL | 2019 | The substrate-bound crystal structure of HPRT (hypoxanthine phosphoribosyltransferase) | https://www.rcsb.org/structure/6D9R | Protein Data Bank, 6D9R |
| Satyshur KA, Dubiel K, Anderson B, Wolak C, Keck JL | 2019 | The (p)ppGpp-bound crystal structure of HPRT (hypoxanthine phosphoribosyltransferase) | https://www.rcsb.org/structure/6D9S | Protein Data Bank, 6D9S |

The following previously published datasets were used:

| Author(s) | Year | Dataset title | Dataset URL | Database and Identifier |
|---|---|---|---|---|
| Zhang N, Gong X, Lu M, Chen X, Qin X, Ge H | 2016 | Crystal structures of Apo and GMP bound hypoxanthine-guanine phosphoribosyltransferase from Legionella pneumophila and the implications in gouty arthritis | https://www.rcsb.org/structure/5esw | Protein Data Bank, 5ESW |
| Zhang N, Gong X, Lu M, Chen X, Qin X, Ge H | 2016 | Crystal structures of Apo and GMP bound hypoxanthine-guanine phosphoribosyltransferase from Legionella pneumophila and the implications in gouty arthritis | https://www.rcsb.org/structure/5esx | Protein Data Bank, 5ESX |
| Halavaty AS, Shuvalova L, Minasov G, Dubrovska I, Peterson SN, Anderson WF | 2009 | 2.06 Angstrom resolution structure of a hypoxanthine-guanine phosphoribosyltransferase (hpt-1) from Bacillus anthracis str. 'Ames Ancestor' | https://www.rcsb.org/structure/3h83 | Protein Data Bank, 3H83 |

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
