## [Decision Letter]

Thank you for submitting your article "Evolution of (p)ppGpp-HPRT regulation through diversification of an allosteric oligomeric interaction" for consideration by *eLife*. Your article has been reviewed by three peer reviewers, one of whom is a member of our Board of Reviewing Editors, and the evaluation has been overseen by Michael Marletta as the Senior Editor.

The reviewers have discussed the reviews with one another and the Reviewing Editor has drafted this decision to help you prepare a revised submission.

Summary:

In this manuscript, the authors investigated the inhibition of HPRT, a key enzyme for the salvage of purine nucleotides, by the bacterial alarmone (p)ppGpp. They purified recombinant HPRT homologs from a broad spectrum of organisms for in vitro analysis, and revealed a significant variation of their affinities to (p)ppGpp. They then performed in-depth studies of pppGpp-sensitive HPRT homologs from Bacillus species using a combination of structural, biophysical and biochemical techniques. Surprisingly, although the binding site of pppGpp almost completely overlaps with that of HPRT substrates, substrate binding leads to the dissociation of HPRT tetramer, which exists in apo- or pppGpp-bound states, into dimers. By comparing sequences, structures, and oligomerization properties of HPRT homologs sensitive or refractory to (p)ppGpp inhibition, the authors concluded that tetramerization of HPRT triggers structural rearrangements to enhance (p)ppGpp binding, and that a dimer-dimer interface supporting tetramerization is the defining feature of (p)ppGpp sensitivity among HPRT homologs. They call this interplay between oligomeric state and ligand affinity "oligomeric allostery". Using ancestral protein reconstruction, the authors also suggested that the emergence of the dimer-dimer interface is likely coupled to the evolution of (p)ppGpp sensitivity.

Although there was enthusiasm for the work, the reviewers collectively raised a number of concerns, some about the work itself and some about the interpretation or presentation of the data. Some of these concerns should be straightforward to address experimentally. Others may be more involved, particularly points 4-5 and the broader issue of whether this mechanism is relevant in vivo given the physiological concentrations of the molecules in question. If the authors feel that all of the issues could be addressed, the reviewers would welcome a revision.

Essential revisions:

1) The authors cited others' work and their own ITC data and claimed that HPRT is not an allosteric enzyme. In fact, apo-HPRT is likely a tetramer at the working concentration for ITC (45 μM) so that the experiment is not expected to reveal cooperativity. Conversely, it is possible that apo-HPRT falls apart into dimers when diluted to ~100 nM for biochemical assays so that the cooperativity for pRpp was not observed. Nonetheless, if pRpp binding dissociates HPRT tetramers into dimers, ITC of HPRT with pRpp should reveal cooperativity, and the [pRpp]-Vi relationship should deviate from the Michaelis-Menten model when HPRT concentration is sufficient to maintain a tetrameric configuration in the apo-form. The authors should present these data as important supporting evidence for their oligomeric allostery model.

2) Throughout the second half of the manuscript, the authors emphasize the significance of tetramerization on tight (p)ppGpp binding, but touched very little on the other side of the allostery, namely, how tetrameric vs dimeric HPRT affects the enzyme activity? For instance, is HPRT made constitutively tetrameric (e.g., crosslinked with disulfides) less active? And can the phylogenetic studies be used not to break a tetrameric HPRT (as is already done) but to make a normally dimeric HPRT tetrameric and thus subject to ppGpp-based regulation?

3) Subsection “(p)ppGpp regulation of HPRT is conserved across bacteria and beyond”: "these results suggest significant inhibition of HPRT under physiological, basal levels of (p)ppGpp" Provided that inhibition by (p)ppGpp is competitive with respect to pRpp, the magnitude of inhibition also depends on the physiological level and Km for pRpp. In other words, it's difficult to say in vivo what the effect of ppGpp will be (especially at low concentrations) unless one also knows the concentration of pRpp and the enzyme's Km for pRpp. Can these values be measured? If not, it may significantly impact what can be concluded about the level of inhibition by ppGpp in vivo.

4) In the end, it's not clear how important the ppGpp-driven stabilization of a tetramer is for inhibition. In the chimera experiment in Figure 4, the authors disrupt tetramerization and the affinity for ppGpp decreases 20-fold. But is this chimera no longer inhibited by ppGpp or does it just take higher levels? As the authors point out, ppGpp levels can get into the mM regime, so how much does a change in affinity from 1 to 20 μm matter in vivo? Related to this, the authors only examine enzyme inhibition in Figure 1 at one concentration: 25 uM. Maybe some of the enzymes, like the *L. pneumophila* enzyme used in creating the chimera in Figure 4, are more fully inhibited at even slightly higher concentrations. We wouldn't expect the authors to examine enzyme inhibition in detail for every organism's HPRT, every mutant, and the chimera in Figure 4, but in a few select cases (especially the chimera and a couple of the orthologs) it seems essential to do so.

5) What are the in vivo consequences of not forming a tetramer in *B. subtilis*, with respect to GTP homeostasis?

6) Ancestral reconstruction

The ancestral reconstruction method used was not correct and needs to be redone. This is a critical problem, as they authors do not establish that the branching pattern of their phylogeny is correct for the deepest nodes (Anc 1-7). If the branching pattern was different, these ancestors simply would not have existed. As a result, it is not clear whether the ancestor was a dimer that became a tetramer (as they claim) or was instead a tetramer that became a dimer.

The solution to these issues would be to do the reconstruction using established methods (see below) using a much larger set of sequences than the 141 chosen. It may be that their conclusions are robust to a better reconstruction, but they must establish this before publication.

A) None of the nodes they reconstructed (Anc 1-7) have bootstrap values above 90. They do not report the support values for these nodes, but I suspect that the support for these nodes is very low (see D below). Even though the posterior probabilities on the reconstruction of ancestral amino acids are high for the sites they identify as important, the authors have not convinced me that these nodes actually existed historically. Maybe the lineage lacking the dimer-dimer motif evolved later, representing a loss of ancestral tetramerization. This possibility cannot be distinguished from the hypothesis they present on the basis of their tree.

B) They constructed their tree using the WAG+G+I substitution model, but then reconstructed their ancestors using the JTT model. This mismatch is not correct and likely changed their reconstructed ancestral amino acid states. This should be re-done using consistent models.

C) They never justified the substitution models they chose (usually done using something like ProtTest or a manual AIC calculation). This should be done.

D) If the branch length unit shown on their tree (Figure 5A) is substitutions per site (as it usually is), their placement of the *S. cerevisiae* protein is highly suspect. By eye, that branch accumulated ~1.5 subs/site, indicating it likely aligned very poorly. All one can see from the tree is that the bootstrap value for this placement is <90 – but likely it is much lower. There are several strategies they could/should employ to get around this. First, build a much larger alignment that includes multiple representative sequences from the outgroup clade. This should better resolve the phylogeny. Second, report support values on all nodes-particularly the nodes that they use for reconstruction studies. If the support for these nodes remains low, strong claims about the order of branching and order of evolution of dimerization and tetramerization cannot be made.

E) They show a branch length label of "0.2" on their tree (Figure 5A), but do not define it in the legend. This should be done.

F) They did not describe what method they used for the reconstruction. They cite Jones, 1992, but this is the JTT substitution model, not the reconstruction method. Presumably they are using a marginal probability reconstruction (e.g. Yang, 1995). This should be stated.

G) To ensure reproducibility, the authors should publish their alignment in the supplement.

7) "Oligomeric Allostery"

The authors spend significant time in the discussion arguing that their work reveals a new phenomenon they call "oligomeric allostery." It does not seem it was ever defined precisely in the text; however, as we understood it, it is a change in oligomeric state that controls ligand specificity. They point out in numerous places that they think is a new and important observation (Discussion section).

We do not think it is new. The authors note that there are many instances in which ligand binding controls oligomeric state (e.g. subsection “Protein oligomerization allosterically alters ligand specificity”). But, from the perspective of linkage thermodynamics, this is equivalent to saying ligand binding controls oligomeric state. The earliest thinking regarding allostery and cooperativity was informed by studies of oligomeric states (Perutz, MWC, etc.).

In our view, this framing hurts the manuscript, but not the underlying study. We think they could recast the work without relying on this dubious premise. The authors would be better served to argue that ligand binding and oligomeric states are often tightly linked to one another. They could then describe their work as a cool way that this interplay enables evolutionary change (in their case, by allowing mutations to accumulate away from a conserved binding pocket to promote a change in specificity).

[Editors' note: further revisions were requested prior to acceptance, as described below.]

Thank you for submitting your article "Evolution of (p)ppGpp-HPRT regulation through diversification of an allosteric oligomeric interaction" for consideration by *eLife*. Your article has been reviewed by three peer reviewers, one of whom is a member of our Board of Reviewing Editors, and the evaluation has been overseen by Michael Marletta as the Senior Editor. The reviewers have opted to remain anonymous.

The reviewers have discussed the reviews with one another and the Reviewing Editor has drafted this decision to help you prepare a revised submission.

The reviewers were generally satisfied with your responses to the initial set of concerns raised. There are, however, a couple of remaining issues that you should address in a revision.

1) The first is that the reviewers were not entirely convinced by the proposed mechanism. If we understood it correctly, you are still proposing a cooperative mechanism but have no data supporting cooperativity. This came out most directly in the revision cover letter, where you suggest that cooperativity would not arise if (1) a single (p)ppGpp is sufficient to lock the tetramer in place and (2) four PRPP molecules are necessary to cause dissociation into a dimer. But this did not make sense to the reviewers – one of them even pieced together a set of linkage equations compatible with your verbal model and explored them on a computer and could not find any parameters that did not display cooperativity in PRPP. Your revision should clarify your proposed mechanism and whether cooperativity is a key feature and if so, what results support that interpretation.

2) It was difficult to assess your revisions as you did not always state whether the individual points had given rise to textual changes or not. Some were obvious, but others less so. In your next revision, please address the following:

a) How is the oligomeric state of each structure assessed? While it is clear that the PRPP + 9- deazaguanine bound form is dimeric from the packing in the supplied PDB file, the tetramer observed in the other two structures could in theory be a result of crystal packing. Usually, a tool like PISA is used to assess the validity of proposed multimers in crystal structures, but the authors make no reference to this. Please confirm that PISA agrees with the conclusions about oligomeric state.

Thank you for including the PISA analysis in the response to reviewers, this is quite clear. I also notice that you now mention PISA in the Materials and methods section. However, I believe it will be of value to the readers to have this information also at the relevant place in the Results section, i.e. whenever you conclude on oligomeric state, I suggest you say that this is supported by PISA analysis.

b) Subsection “(p)ppGpp binds the conserved active site of HPRT and closely mimics substrate binding”. Have the authors considered that the pppGpp preparation could contain a ppGpp contamination that crystallised with the protein? Would it be possible to check for this?

Could the authors please indicate whether this has resulted in any changes to the manuscript?

c) Subsection “(p)ppGpp prevents PRPP-induced dissociation of HPRT dimer-of-dimers” and Figure 3. What is the reason for excluding 9-deazaguanine during gel filtration and cross- linking? The comparison to the crystal structure would be more direct if both substrates were present. Is it possible that the oligomeric state of the protein only in the presence of PRPP is different?

Could the authors please indicate whether this has resulted in any changes to the manuscript? In this case I would suggest that you include this additional experiment as supplementary data in case other readers wonder.

d) Subsection “Dimer-dimer interaction allosterically positions loop II for potentiated (p)ppGpp binding”. Can the authors exclude that the structural variations observed in loop II to some extent are induced by crystal contacts?

Did this give rise to textual changes?

---

## [Author Response]

Essential revisions:1) The authors cited others' work and their own ITC data and claimed that HPRT is not an allosteric enzyme. In fact, apo-HPRT is likely a tetramer at the working concentration for ITC (45 μM) so that the experiment is not expected to reveal cooperativity. Conversely, it is possible that apo-HPRT falls apart into dimers when diluted to ~100 nM for biochemical assays so that the cooperativity for pRpp was not observed. Nonetheless, if pRpp binding dissociates HPRT tetramers into dimers, ITC of HPRT with pRpp should reveal cooperativity, and the [pRpp]-Vi relationship should deviate from the Michaelis-Menten model when HPRT concentration is sufficient to maintain a tetrameric configuration in the apo-form. The authors should present these data as important supporting evidence for their oligomeric allostery model.

This is a very good point. To address this question, we conducted ITC between PRPP and HPRT (~45 μM). The isotherm fit a single site binding model without observed cooperativity for PRPP binding (with a K_d_ ~ 7 μM, new data added as Figure 4 —figure supplement 5). Because we have already observed with dynamic light scattering at 175 μM of HPRT that PRPP still induces tetramer to dimer formation (Table 3), we expect at the 45 μM concentration used for ITC, PRPP would also induce tetramer to dimer transition. Thus, at high HPRT concentrations sufficient to maintain a tetrameric configuration in the apo form, we have not observed cooperativity for PRPP binding despite its ability to induce the tetramer to dimer transition. We assume that the [PRPP]-Vi relationship will not deviate from the Michaelis-Menten model at HPRT concentrations sufficient to maintain a tetrameric configuration in the apo-form.

We propose that the lack of cooperativity is best explained by PRPP having to associate with all four binding sites before it induces the tetramer to dimer transition. This would also explain why (p)ppGpp binding can lock the HPRT as a tetramer despite addition of high concentrations of PRPP – one (p)ppGpp binding may be sufficient to prevent dimeric HPRT formation.

Finally, we reasoned that our original ‘oligomeric allostery’ model where the enzyme is not allosteric is confusing. To remove this confusion, we have removed (as suggested by reviewers in comment 6) the term ‘oligomeric allostery’.

2) Throughout the second half of the manuscript, the authors emphasize the significance of tetramerization on tight (p)ppGpp binding, but touched very little on the other side of the allostery, namely, how tetrameric vs dimeric HPRT affects the enzyme activity?

We found that naturally occurring dimeric HPRT homologs have similar activity compared to the tetrameric *B. anthracis* Hpt-1 and *B. subtilis* HPRT (Table 4). Between the naturally occurring homologs, the biggest difference as far as we can perceive, is their ability to be regulated by (p)ppGpp.

In addition, we found that the majority of the dimeric homologs also tend to have lower K_m_ values for PRPP. This suggests that the evolution of the oligomer is not just affecting the affinity for (p)ppGpp but also for PRPP, and the dimer favors PRPP binding but the tetramer favors (p)ppGpp binding, which should also play a role in their reduced inhibition by (p)ppGpp. This point has been elaborated in the Discussion section.

For instance, is HPRT made constitutively tetrameric (e.g., crosslinked with disulfides) less active? And can the phylogenetic studies be used not to break a tetrameric HPRT (as is already done) but to make a normally dimeric HPRT tetrameric and thus subject to ppGpp-based regulation?

To address this comment, we constructed a presumably constitutively tetrameric HPRT with a Y117C variant. This variant had significantly lower activity than WT (~6% of the Vmax). Also, while this variant bound pppGpp as well as the WT HPRT, it was nearly resistant to high concentrations of competing PRPP. This suggests that PRPP is unable to bind and dissociate a locked tetramer. These data are now shown as Figure 4—figure supplement 4.

We have also attempted to make a normally dimeric HPRT a tetramer. We replaced interface residues of the dimeric *L. pneumophila* HPRT with corresponding residues from *B. subtilis* HPRT. However, this variant was insoluble, perhaps due to improper folding. Despite the strength of the phylogenetic analyses, the HPRT homologs are too sequence diverse to pinpoint all compensatory changes.

3) Subsection “(p)ppGpp regulation of HPRT is conserved across bacteria and beyond”: "these results suggest significant inhibition of HPRT under physiological, basal levels of (p)ppGpp" Provided that inhibition by (p)ppGpp is competitive with respect to pRpp, the magnitude of inhibition also depends on the physiological level and Km for pRpp. In other words, it's difficult to say in vivo what the effect of ppGpp will be (especially at low concentrations) unless one also knows the concentration of pRpp and the enzyme's Km for pRpp. Can these values be measured? If not, it may significantly impact what can be concluded about the level of inhibition by ppGpp in vivo.

We thank the reviewer to remind us to clarify this point. The regulation by basal level is highly relevant in vivo.

1) Physiological PRPP levels in bacteria are roughly 0.1 – 1 mM (Bennett et al., (2009); Jensen, (1983)). In *B. subtilis*, the PRPP level is also in this range (0.2 – 1 mM) (Berlin and Stadtman, (1966)). All inhibition assays were performed with 1 mM PRPP, the high end of PRPP concentration. At this concentration, (p)ppGpp at lower concentrations (25 μM) can still strongly inhibit HPRT activity (Figure 2E). On the other hand, for the constitutive dimeric homologs, at 1 mM PRPP, (p)ppGpp can no longer inhibit HPRT (Figure 2E). We now add this information in the Discussion section.

2) We measured K_m_ for PRPP for *B. subtilis* HPRT and *B. anthracis* Hpt-1 (166 μm and 112 uM, data shown in Table 4). Most bacterial HPRTs have a K_m_ for PRPP in the range of 50 – 200 μM (Guddat et al., (2003); Zhang et al., (2016)). On the other hand, K_d_ values for pppGpp typically range from 0.1-10 μM. Both of these parameters are within the physiological range of PRPP and (p)ppGpp.

3) We performed enzymatic assays showing that (p)ppGpp is a competitive inhibitor of HPRT with its substrate PRPP.

4) In the end, it's not clear how important the ppGpp-driven stabilization of a tetramer is for inhibition. In the chimera experiment in Figure 4, the authors disrupt tetramerization and the affinity for ppGpp decreases 20-fold. But is this chimera no longer inhibited by ppGpp or does it just take higher levels? As the authors point out, ppGpp levels can get into the mM regime, so how much does a change in affinity from 1 to 20 μm matter in vivo?

This is a valid point that we are now able to clarify during revision. Here, it important to point out that regulation by (p)ppGpp is both important at basal level and at the induced level (e.g. 20-fold higher). In fact, basal level of (p)ppGpp has been shown to be very important for physiology, however, most of the (p)ppGpp biochemistry only showed that induced levels of (p)ppGpp regulate its enzymatic targets (i.e. upon starvation or other stress). The key to HPRT regulation is basal level (p)ppGpp. Our previous experiment (Kriel et al., 2012) showed that basal levels of (p)ppGpp are important for GTP homeostasis in vivo, protecting cells by inhibiting the purine salvage pathway at a basal (p)ppGpp concentration that is in strong contrast to the induced sub-millimolar or millimolar concentrations. In the context studied in Kriel et al., guanine is added as a source of purine in cells supplemented with both glucose and amino acids. In wild type cells grown with guanine, (p)ppGpp remained at its basal level (estimated to be only ~30 μM), yet this level of (p)ppGpp prevents excess GTP production through purine salvage and maintains homeostasis. Therefore, our analysis is highly relevant at least in the case of *Bacillus* species.

To translate the importance of a change in affinity from 1 to 20 μM, at a physiological concentration of PRPP (1 mM, assuming PRPP K_m_ = 0.1 mM), a K_i_ of 1 μM for pppGpp will result in ~75% inhibition at 30 μM pppGpp. On the other hand, a K_i_ of 20 μM will result in ~12% inhibition at 30 μM pppGpp. Even upon (p)ppGpp inducing to 0.5 mM, a K_i_ of 20 μM will result in ~70% inhibition of HPRT activity. Thus, lowering the affinity for pppGpp by 20-fold would mean that it is no longer an effective competitor of PRPP, especially at basal (p)ppGpp, and would likely keep the enzyme active.

Related to this, the authors only examine enzyme inhibition in Figure 1 at one concentration: 25 uM. Maybe some of the enzymes, like the L. pneumophila enzyme used in creating the chimera in Figure 4, are more fully inhibited at even slightly higher concentrations. We wouldn't expect the authors to examine enzyme inhibition in detail for every organism's HPRT, every mutant, and the chimera in Figure 4, but in a few select cases (especially the chimera and a couple of the orthologs) it seems essential to do so.

In the revised version, IC_50_ curves of multiple orthologs, including *C. burnetii*, are shown in Figure 2E. For *C. burnetii*, IC_50_ > 100 μM, much stronger than the 10 μM IC_50_ for *B. subtilis*.

5) What are the in vivo consequences of not forming a tetramer in B. subtilis, with respect to GTP homeostasis?

We performed several experiments to examine the in vivo consequences of weakening the affinity of HPRT for (p)ppGpp with respect to GTP homeostasis. We first attempted to introduce the (p)ppGpp-refractory *C. burnetii hprT* into *B. subtilis* (Δ*hprT* + *amyE*∷*cbuhprT* that expresses *C. burnetii* HPRT from an IPTG inducible promoter). However, the homolog has a weak expression in *B. subtilis* making it unable to utilize guanine efficiently (shown by a lack of growth rate increase upon guanine addition). We next constructed a codon-optimized *C. burnetii hprT*, but this allele is unable to transform into *B. subtilis*, presumably because of an unidentified toxicity.

Eventually, we found a way to demonstrate the physiological importance of basal (p)ppGpp regulation of guanine salvage by introducing a *E. coli* guanine salvage enzyme XGPRT into *B. subtilis*. We biochemically characterized this enzyme to show that its ability to catalyze PRPP and guanine conversion to GMP is inhibited by (p)ppGpp at IC_50_ of 45 μM. In other words, this enzyme is inhibited by induced, but not basal level of (p)ppGpp (Figure 1B). When we introduced this construct into an *ecgmk* background (a background where *B. subtilis* GMK which is strongly inhibited by (p)ppGpp is replaced by *E. coli* GMK that is resistant to (p)ppGpp), we are able to observe a much higher concentration of GTP (4-fold higher) upon addition of guanosine (Figure 1C). Therefore, we are able to verify in vivo consequences of an HPRT that is inhibited only by induced (p)ppGpp, with respect to GTP homeostasis.

6) Ancestral reconstructionThe ancestral reconstruction method used was not correct and needs to be redone. This is a critical problem, as they authors do not establish that the branching pattern of their phylogeny is correct for the deepest nodes (Anc 1-7). If the branching pattern was different, these ancestors simply would not have existed. As a result, it is not clear whether the ancestor was a dimer that became a tetramer (as they claim) or was instead a tetramer that became a dimer.The solution to these issues would be to do the reconstruction using established methods (see below) using a much larger set of sequences than the 141 chosen. It may be that their conclusions are robust to a better reconstruction, but they must establish this before publication.

We are thankful to the reviewers for their helpful criticism of the phylogenetic methods. We re-performed the phylogenetic analysis with an expanded 430 HPRT sequences with the specific issues raised below in mind. With the new analysis, we still obtain a phylogenetic tree that branches into two broad lineages: one sensitive to basal (p)ppGpp and one insensitive to basal (p)ppGpp. The ancestor to the (p)ppGpp-sensitive clade has a relatively low bootstrap value (55-60), indicating low confidence in this ancestor, as pointed out in issue A below. The key is that the residues we had previously identified as being important within each branch are still found to be important with the new analysis – one branch contains residues associated with dimeric HPRTs and the other does not. The Anc1 does not have residues associated with dimeric HPRTs, and it likely contains the Lys81 associated with the (p)ppGpp-regulated clade, suggesting that it is a tetramer.

The lower confidence in these branches inspired us to analyze another enzyme differentially regulated by (p)ppGpp: GMK, which has a much higher sequence identity across organisms than HPRT. Even with fewer sequences, this resulted in a more robust tree that also differentiates (p)ppGpp-sensitive and insensitive GMKs into two broad clades and suggests that the ancestral GMK was also regulated by (p)ppGpp.

A) None of the nodes they reconstructed (Anc 1-7) have bootstrap values above 90. They do not report the support values for these nodes, but I suspect that the support for these nodes is very low (see D below). Even though the posterior probabilities on the reconstruction of ancestral amino acids are high for the sites they identify as important, the authors have not convinced me that these nodes actually existed historically. Maybe the lineage lacking the dimer-dimer motif evolved later, representing a loss of ancestral tetramerization. This possibility cannot be distinguished from the hypothesis they present on the basis of their tree.

This is a very good point and is addressed above.

B) They constructed their tree using the WAG+G+I substitution model, but then reconstructed their ancestors using the JTT model. This mismatch is not correct and likely changed their reconstructed ancestral amino acid states. This should be re-done using consistent models.

We used ProtTest to identify the best substitution model with the lowest AIC score as LG+G+I. Ancestral reconstruction was performed with the same model.

C) They never justified the substitution models they chose (usually done using something like ProtTest or a manual AIC calculation). This should be done.

See (B) above.

D) If the branch length unit shown on their tree (Figure 5A) is substitutions per site (as it usually is), their placement of the S. cerevisiae protein is highly suspect. By eye, that branch accumulated ~1.5 subs/site, indicating it likely aligned very poorly. All one can see from the tree is that the bootstrap value for this placement is <90 – but likely it is much lower. There are several strategies they could/should employ to get around this. First, build a much larger alignment that includes multiple representative sequences from the outgroup clade. This should better resolve the phylogeny. Second, report support values on all nodes-particularly the nodes that they use for reconstruction studies. If the support for these nodes remains low, strong claims about the order of branching and order of evolution of dimerization and tetramerization cannot be made.

This branch length unit is the substitutions per site. We included more sequences, particularly in the outgroup clade, to get a better alignment. All bootstrap values are now reported, especially for the ancestral states.

E) They show a branch length label of "0.2" on their tree (Figure 5A), but do not define it in the legend. This should be done.

The branch length label has been defined in the legend.

F) They did not describe what method they used for the reconstruction. They cite Jones, 1992, but this is the JTT substitution model, not the reconstruction method. Presumably they are using a marginal probability reconstruction (e.g. Yang, 1995). This should be stated.

Correct, the reconstruction method was according to Yang, 1995. This has been cited in the Results section and described in the Materials and methods section.

G) To ensure reproducibility, the authors should publish their alignment in the supplement.

The alignments have been included as Source Data files (Figure 6—source data 1 and Figure 7—source data 1).

7) "Oligomeric Allostery"The authors spend significant time in the discussion arguing that their work reveals a new phenomenon they call "oligomeric allostery." It does not seem it was ever defined precisely in the text; however, as we understood it, it is a change in oligomeric state that controls ligand specificity. They point out in numerous places that they think is a new and important observation (Discussion section).We do not think it is new. The authors note that there are many instances in which ligand binding controls oligomeric state (e.g. subsection “Protein oligomerization allosterically alters ligand specificity”). But, from the perspective of linkage thermodynamics, this is equivalent to saying ligand binding controls oligomeric state. The earliest thinking regarding allostery and cooperativity was informed by studies of oligomeric states (Perutz, MWC, etc.).In our view, this framing hurts the manuscript, but not the underlying study. We think they could recast the work without relying on this dubious premise. The authors would be better served to argue that ligand binding and oligomeric states are often tightly linked to one another. They could then describe their work as a cool way that this interplay enables evolutionary change (in their case, by allowing mutations to accumulate away from a conserved binding pocket to promote a change in specificity).

We thank the reviewers for their feedback on this point and for the proposed reinterpretation. We have altered our model and modified the text at multiple points to incorporate these ideas.

[Editors' note: further revisions were requested prior to acceptance, as described below.]

The reviewers were generally satisfied with your responses to the initial set of concerns raised. There are, however, a couple of remaining issues that you should address in a revision.1) The first is that the reviewers were not entirely convinced by the proposed mechanism. If we understood it correctly, you are still proposing a cooperative mechanism but have no data supporting cooperativity. This came out most directly in the revision cover letter, where you suggest that cooperativity would not arise if (1) a single (p)ppGpp is sufficient to lock the tetramer in place and (2) four PRPP molecules are necessary to cause dissociation into a dimer. But this did not make sense to the reviewers – one of them even pieced together a set of linkage equations compatible with your verbal model and explored them on a computer and could not find any parameters that did not display cooperativity in PRPP. Your revision should clarify your proposed mechanism and whether cooperativity is a key feature and if so, what results support that interpretation.

We thank the reviewers for the opportunity to clarify this point. We do not think that cooperativity is a key feature of our model. We can confidently say that variation in the dimer-dimer interface influences the selectivity for substrate versus inhibitor. The potential cooperativity of PRPP binding (or lack thereof) does not change our findings that the dimer prefers PRPP over (p)ppGpp and the tetramer prefers (p)ppGpp over PRPP.

We thank the reviewers for pointing out that cooperativity would still arise if single (p)ppGpp is sufficient to lock the tetramer in place or four PRPP molecules are necessary to cause dissociation into a dimer. We agree that cooperativity is likely detectable under certain conditions. However, in the conditions we tested, within physiological ranges of (p)ppGpp and PRPP, we did not detect cooperativity. Our literature search of this extensively characterized enzyme did not find observation of cooperativity of native HPRT. On the other hand, kinetic cooperativity between PRPP and HPRT has been reported in mutant human HPRTs (Lightfoot et al., 1994; Balendiran et al., 1999). Thus, observing this cooperativity requires altering the native enzyme.

In light of these investigations, we revised the argument about the physiological relevance of cooperativity in the main text. We feel that the rest of the manuscript would be clear and impactful without the argument of cooperativity related to PRPP.

2) It was difficult to assess your revisions as you did not always state whether the individual points had given rise to textual changes or not. Some were obvious, but others less so. In your next revision, please address the following:a) How is the oligomeric state of each structure assessed? While it is clear that the PRPP + 9- deazaguanine bound form is dimeric from the packing in the supplied PDB file, the tetramer observed in the other two structures could in theory be a result of crystal packing. Usually, a tool like PISA is used to assess the validity of proposed multimers in crystal structures, but the authors make no reference to this. Please confirm that PISA agrees with the conclusions about oligomeric state.Thank you for including the PISA analysis in the response to reviewers, this is quite clear. I also notice that you now mention PISA in the Materials and methods section. However, I believe it will be of value to the readers to have this information also at the relevant place in the Results section, i.e. whenever you conclude on oligomeric state, I suggest you say that this is supported by PISA analysis.

Thanks! We added these statements in subsection “p)ppGpp prevents PRPP-induced dissociation of HPRT dimer-of-dimers”.

b) Subsection “(p)ppGpp binds the conserved active site of HPRT and closely mimics substrate binding”. Have the authors considered that the pppGpp preparation could contain a ppGpp contamination that crystallised with the protein? Would it be possible to check for this?Could the authors please indicate whether this has resulted in any changes to the manuscript?

We have added this observation in subsection “(p)ppGpp binds the conserved active site of HPRT and closely mimics substrate binding”.

c) Subsection “(p)ppGpp prevents PRPP-induced dissociation of HPRT dimer-of-dimers” and Figure 3. What is the reason for excluding 9-deazaguanine during gel filtration and cross- linking? The comparison to the crystal structure would be more direct if both substrates were present. Is it possible that the oligomeric state of the protein only in the presence of PRPP is different?Could the authors please indicate whether this has resulted in any changes to the manuscript? In this case I would suggest that you include this additional experiment as supplementary data in case other readers wonder.

The data have been added as Figure 4—figure supplement 4, and changes have been made to the Figure 4 legend.

d) Subsection “Dimer-dimer interaction allosterically positions loop II for potentiated (p)ppGpp binding”. Can the authors exclude that the structural variations observed in loop II to some extent are induced by crystal contacts?Did this give rise to textual changes?

This has been clarified in subsection “p)ppGpp prevents PRPP-induced dissociation of HPRT dimer-of-dimers” and the figure has been added as Figure 4—figure supplement 1.